# Human endogenous retroviruses form a reservoir of T cell targets in hematological cancers

Sunil Kumar Saini[1,12], Andreas Due Ørskov [2,3,12], Anne-Mette Bjerregaard [1,12], Ashwin Unnikrishnan [4,5],
Staffan Holmberg-Thydén [1,6], Annie Borch[1], Kathrine Valentini Jensen[1], Govardhan Anande[4,5],
Amalie Kai Bentzen[1], Andrea Marion Marquard[1], Tripti Tamhane[1], Marianne Bach Treppendahl[2],
Anne Ortved Gang[6], Inge Høgh Dufva[6], Zoltan Szallasi [7,8], Nicola Ternette [9], Anders Gorm Pedersen [7],
Aron Charles Eklund[7], John Pimanda [4,5,10], Kirsten Grønbæk[2,3,11] & Sine Reker Hadrup [1✉]

Human endogenous retroviruses (HERV) form a substantial part of the human genome, but mostly remain transcriptionally silent under strict epigenetic regulation, yet can potentially be reactivated by malignant transformation or epigenetic therapies. Here, we evaluate the potential for T cell recognition of HERV elements in myeloid malignancies by mapping transcribed HERV genes and generating a library of 1169 potential antigenic HERV-derived peptides predicted for presentation by 4 HLA class I molecules. Using DNA barcode-labeled MHC-I multimers, we find CD8$^+$ T cell populations recognizing 29 HERV-derived peptides representing 18 different HERV loci, of which HERVH-5, HERVW-1, and HERVE-3 have more profound responses; such HERV-specific T cells are present in 17 of the 34 patients, but less frequently in healthy donors. Transcriptomic analyses reveal enhanced transcription of the HERVs in patients; meanwhile DNA-demethylating therapy causes a small and hetero-geneous enhancement in HERV transcription without altering T cell recognition. Our study thus uncovers T cell recognition of HERVs in myeloid malignancies, thereby implicating HERVs as potential targets for immunotherapeutic therapies.

[1] Department of Health Technology, Section of Experimental and Translational Immunology, Technical University of Denmark, Kongens Lyngby, Denmark. [2] Department of Haematology, Rigshospitalet, Copenhagen University Hospital, Copenhagen, Denmark. [3] Biotech Research and Innovation Centre (BRIC), University of Copenhagen, Copenhagen, Denmark. [4] Adult Cancer Program, Lowy Cancer Research Centre, UNSW, Sydney, NSW 2052, Australia. [5] Prince of Wales Clinical School, UNSW, Sydney, NSW 2052, Australia. [6] Department of Haematology, Herlev Hospital, Copenhagen University Hospital, Herlev, Denmark. [7] Department of Health Technology, Section of Bioinformatics, Technical University of Denmark, Kongens Lyngby, Denmark. [8] Computational Health Informatics Program (CHIP), Boston Children's Hospital, Harvard Medical School, Boston, MA, USA. [9] Nuffield Department of Medicine, University of Oxford, Oxford, UK. [10] Haematology Department, South Eastern Area Laboratory Services, Prince of Wales Hospital, Randwick, NSW 2031, Australia. [11] Novo Nordisk Foundation Center for Stem Cell Biology (DanStem), University of Copenhagen, Copenhagen, Denmark. [12] These authors contributed equally: Sunil Kumar Saini, Andreas Due Ørskov, Anne-Mette Bjerregaard. ✉email: sirha@dtu.dk

Human endogenous retroviruses (HERVs) are inherited genetic germline elements derived from exogenous retroviral infections throughout the evolution of the human genome, and account for ~8% of our genome[1,2]. The majority of HERVs are defective due to evolutionarily acquired disruption or silencing mutations[2,3]. Hence, no infectious activity remains from such HERVs, but they may still be recognized as viral components by our immune system. Interestingly, elevated HERV expression has been associated with a variety of autoimmune disorders and cancers, although the causative roles and pathogenicity of HERVs have not been clarified[4–13].

Given the potential mutagenic consequences of replications and re-insertions of HERVs, human cells have developed different mechanisms to suppress HERV transcription and retrotransposition[2,3,14]. Hence, epigenetic mechanisms serve to transcriptionally silence the HERV-derived elements through DNA methylation and/or repressive histone modifications[15]. Previous work has shown that the loss of DNA methylation mediated by DNA-demethylating therapy (hypomethylating agents [HMAs]), leads to an upregulation of single- and double-stranded HERV transcripts in human cancer cell lines, revealing a functional role of DNA methylation in repressing HERV expression[16–18]. Recently, we could also show that this occurs in patients[19]. Moreover, malignant transformation may by itself induce the expression of HERVs due to the global DNA hypomethylation observed in cancer genomes[12,16,17,20].

The HMAs, 5-azacytidine (AZA) and decitabine, have been shown to be effective in patients with certain hematological malignancies, such as higher-risk myelodysplastic syndromes (MDS), chronic myelomonocytic leukemia (CMML), and acute myeloid leukemia (AML)[21,22]. Interestingly, a key anti-tumor effect of HMAs may rely on the activation of innate immunological mechanisms induced by HERV derepression[19,23,24]. Thus, to take advantage of this biological phenomenon, and to further boost immune recognition of malignant cells, several trials have opened, combining HMAs and immune checkpoint inhibitors[25]. In this study, we aim to investigate if HERV elements may form a reservoir of antigens that can at large scale be targeted through T

cell recognition. Selected HERV-derived epitopes have been described as tumor-associated antigens, which can promote tumor-specific recognition by T cells in vivo[26–33].

Here we include a large library of HERV-derived peptides to determine if T cell-directed HERV recognition is present in patients with cancers of low mutational burden. To investigate the in vivo existence of T cells able to recognize HERV-derived peptide-antigens, we utilize a DNA barcode-labeled MHC-I multimer technology recently established by our laboratory. Our HERV library included HLA-binding ligands, covering 4 different HLA class I molecules, predicted from the sequences of 66 HERVs that were previously described to potentially retain translational activity in human tissues[34]. We analyze 34 patients and 27 healthy donors for T cell recognition of 1169 different HERV-derived peptide-MHCs (pMHC) and find that HERV-directed T cells are enriched in patients with myeloid malignancies. Our results implicate that T cell recognition of HERVs could be leveraged in future immunotherapeutic strategies.

## Results

**Prediction of HERV-derived T cell antigens.** To examine whether HERVs can function as a source of cancer-specific antigens and potentially drive T cell activation in patients with myeloid malignancies, we first attempted to identify HERV-derived peptides potentially presented on cell-surface MHC class I molecules. We focused on 66 systematically annotated HERV loci, previously reported to be transcribed, either in normal or diseased human cells[34] (Supplementary Data 1). The 66 HERVs originate from 17 different groups, annotated according to Repbase, and of these the HERVK group has the highest representation (39%). The 66 HERVs are generally transcribed at very low levels in the majority of the healthy tissues as well as in cancers (Supplementary Fig. 1). The exception to this is ERVV-2, ERVV1, ERVK-6, ERVFRD-2, ERVFH21-1, which are all highly transcribed across both normal and cancerous tissue. Some HERVs are also tumor and tissue specific, for example, ERVH-5, ERVH48-1 and ERVE-4 shown by Rooney et al. to be tumor specific[12].

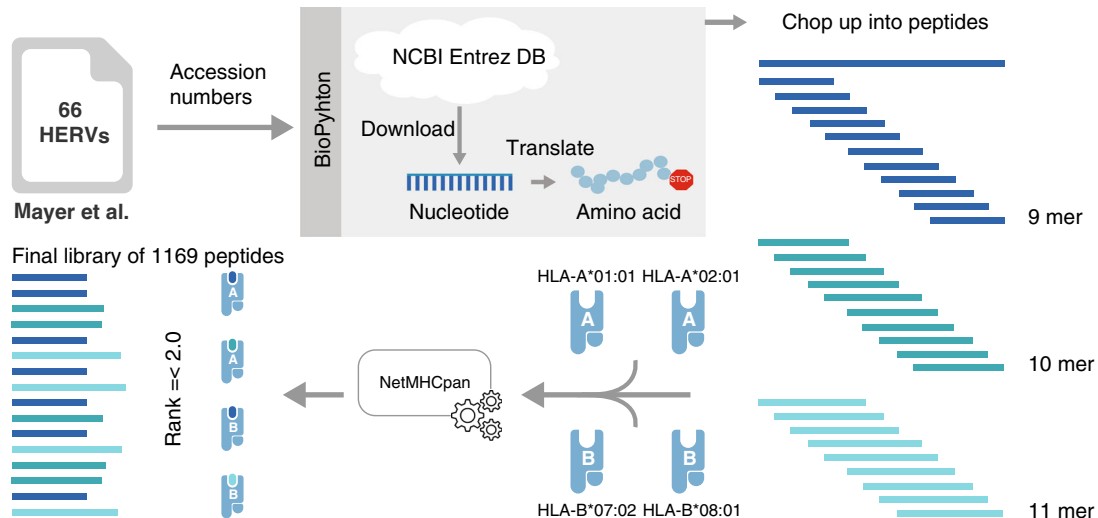

**Fig. 1 Schematic of HERV-derived peptides selected for T cell analysis.** Accession numbers for known transcribed HERVs published by Mayer et al. were used to download the reported nucleotide sequences from the National Center for Biotechnology Information's (NCBIs) Entrez database (DB). The sequences were translated into amino acids until a stop codon occurred. Thereafter, the sequences were chopped into 9-, 10-, and 11mer peptides. Binding of all extracted peptides to the four most common Caucasian HLA alleles (HLA-A*01:01, -A*02:01, -B*07:02, and -B*08:01) was predicted using NetMHCpan 2.8. The final library consists of 1169 peptides from 49 of the 66 HERVs, all with a predicted binding percentile rank score of 2 or below to any of the four HLA alleles.

We downloaded the nucleotide sequences of the 66 HERVs, translated them into amino acids, and extracted 9–11mer peptides predicted to bind to the four most common Caucasian HLA class I alleles: HLA-A*01:01, HLA-A*02:01, HLA-B*07:02, and HLA-B*08:01 (Fig. 1). Of the 66 HERVs, 57 could be translated into amino acid sequences that were long enough to include 9–11mer peptides, and 49 of the HERVs included binders to at least one of the four HLA alleles. The total number of extracted HERV-derived 9–11mer pMHCs was 61,380 and allowed a prediction frequency of 1.9%. Using the recommended pMHC-binding rank score ≤2[35] for NetMHCpan 2.8 on all four HLAs resulted in 1173 peptides predicted for presentation by the MHC I molecules. Out of these peptides, 1169 were successfully synthesized and included in the final library to examine T cell reactivity (199 for HLA-A*01:01, 237 for HLA-A*02:01, 449 for HLA-B*07:02, and 284 for HLA-B*08:01, Supplementary Data 2). Of these, 1036 peptides were unique binders to a given HLA, and the remaining were found to bind two or more of the selected HLA molecules.

**T cell recognition of HERV-derived elements is present in myeloid malignancies**. To investigate the presence of CD8+ T cell populations reactive to HERV-derived peptides as a consequence of the malignant transformation and to further evaluate if HMA therapy impacts the T cell reactivity to our selected group of HERVs, we analyzed samples from 34 patients before and after AZA treatment (MDS, AML, and CMML; Supplementary Table 1). Patient samples collected before and after AZA treatment were included with lymphocytes derived either from peripheral blood mononuclear cells (PBMCs) ($n = 22$, Danish patients) or bone marrow mononuclear cells (BMMCs) ($n = 12$, Australian patients). HERV-specific T cell responses were analyzed in up to four post-treatment samples per patient, ranging from time points at the end of treatment cycle 1 to the end of treatment cycle 6, depending on sample availability. For a comprehensive comparison, PBMCs from 27 healthy donors were included in the analysis.

T cell recognition of the HERV-derived 1,169 peptides was evaluated using a DNA barcode-labeled pMHC-I-multimer-based multiplex technology that enables the analysis of T cell reactivity against large peptide libraries in a single biological sample[36]. First, each pMHC multimer was tagged with an individual DNA barcode. Then pMHC multimer reagents were selected and pooled according to the patients' HLA type, and CD8+ T cells were tested for their ability to recognize (bind to) such pMHC multimers carrying HERV-derived peptides. In addition to the peptide library derived from HERV elements, we also included a library of 67 previously reported cancer testis antigen (CTA)-derived T cell epitopes (Supplementary Data 3)[37]. CTAs are similarly to HERVs regulated by epigenetic mechanisms and re-expression has been demonstrated following malignant transformation and DNA-demethylating therapy[38]. Moreover, 19 T cell epitopes from common viruses, such as Epstein-Barr virus (EBV), cytomegalovirus (CMV), and influenza (Flu) virus, were included as positive controls and as a measure of the general immune status (Supplementary Data 4).

pMHC-reactive CD8+ T cells were identified after sorting of MHC multimer-binding T cells using fluorescence-activated cell sorting (FACS) based on phycoerythrin (PE) fluorescence intensity and the composition of the associated DNA barcodes was retrieved by DNA amplification sequencing. Sequencing data were processed with the software package 'Barracoda' (http://www.cbs.dtu.dk/services/Barracoda, see "Methods"). T cell populations were identified based on DNA barcode enrichment (FDR < 0.1%) in the sorted population compared to the full pMHC

library used for T cell staining (Supplementary Fig. 2). We detected a substantial increase in T cell reactivity toward HERV-peptides in patient samples as compared to healthy donors. Out of 34 patients, 17 had one or more HERV-specific CD8+ T cell populations as compared to only 4 out of 27 healthy donors (Fig. 2, red squares). The number of HERV-reactive T cell populations in patient samples was independent of the source of the samples; peripheral blood or bone marrow (Supplementary Fig. 3a). We identified 29 unique HERV-derived epitopes capable of raising a T cell response across the four tested HLAs (4, HLA-A*01:01; 10, HLA-A*02:01; 10, HLA-B*07:02; and 5, HLA-B*08:01) (Fig. 2a–d). These 29 peptides derived from 18 different HERV loci. Of the 29 peptides, 23 were recognized only in patients (pre- and post-AZA treatment), five in healthy donors, and only one specificity was shared between a healthy donor and a patient sample (Table 1 and Fig. 2e). A large fraction of the HERV loci included in this study were found to be immunogenic in patients (18 out of 49; 37%); each giving rise to at least one recognized T cell epitope.

Applying a T cell reactivity score calculated based on the total peptides recognized by T cells in all analyzed patients out of total peptides derived from each HERV locus identifies HERVH-5 as the most immunogenic of the 49 HERVs analyzed, followed by ERVW-1 and ERVE-3 (Fig. 2f). Importantly, T cell reactivity was independent of the peptide-library size as some of the HERVs with very small peptide libraries showed a relatively high T cell reactivity and vice versa (Fig. 2f). Among individual HERVs, a maximum of five epitopes was derived from one single HERV (ERVFRD-1), whereas most of the remaining epitopes derive from several members of the HERVK family (Table 1). Moreover, several of the HERV-derived epitopes served as shared antigens based on the presence of epitope-reactive T cells in more than one patient: HLA-A*01:01-WTGTCTIGY, two patients; HLA-A*02:01-FLLTSFTTGRV, three patients; HLA-A*02:01-CLISILVSSL, two patients; and HLA-B*07:02- RPRVLRLISPR, three patients (Fig. 2a–d).

In addition to HERVs, T cells reactive to several previously described epitopes from CTAs were identified in six of the patients, as compared to only two in healthy donors (Fig. 2a–d). In the patient samples, we identified T cells reactive to 13 CTAs, out of which 11 were unique to the patient cohort (representing NY-ESO-1, MAGE10, MAGEC1, Cyclin A1, TAG-1, CDCA1, RHAMM, TRAG-3, and PRAME). Furthermore, a large number of viral antigen-reactive T cells were detected for some of the common viral antigens (CMV, EBV, and Flu) in both patients and healthy donors (Fig. 2a–d).

**HERV-specific T cells are significantly enriched in patients**. Based on similar cohort sizes of patients ($n = 34$) and healthy donors ($n = 27$), we compared the proportions of individuals with HERV-reactive T cells in the two groups. The proportion of individuals with HERV-reactive T cell populations was significantly higher in the patient cohort compared to the healthy individuals (posterior probability that proportion is higher: 99% for pre-AZA samples and 97% for post-AZA samples; Fig. 3a). Similarly, the number of different HERV peptides recognized by T cell populations in individual samples was also significantly higher in the patient cohort (before treatment) compared to healthy donors (Fig. 3b), again showing a malignancy-driven enrichment of HERV-reactive T cells in the patient groups. To ensure that this result was not biased by individual HLA profiles (which could be unequally distributed between patients and controls), we used regression modeling to estimate HERV-specific T cell reactivity corrected for HLA influence (Supplementary Fig. 4a, b). Based on this model, both pre-AZA and post-AZA

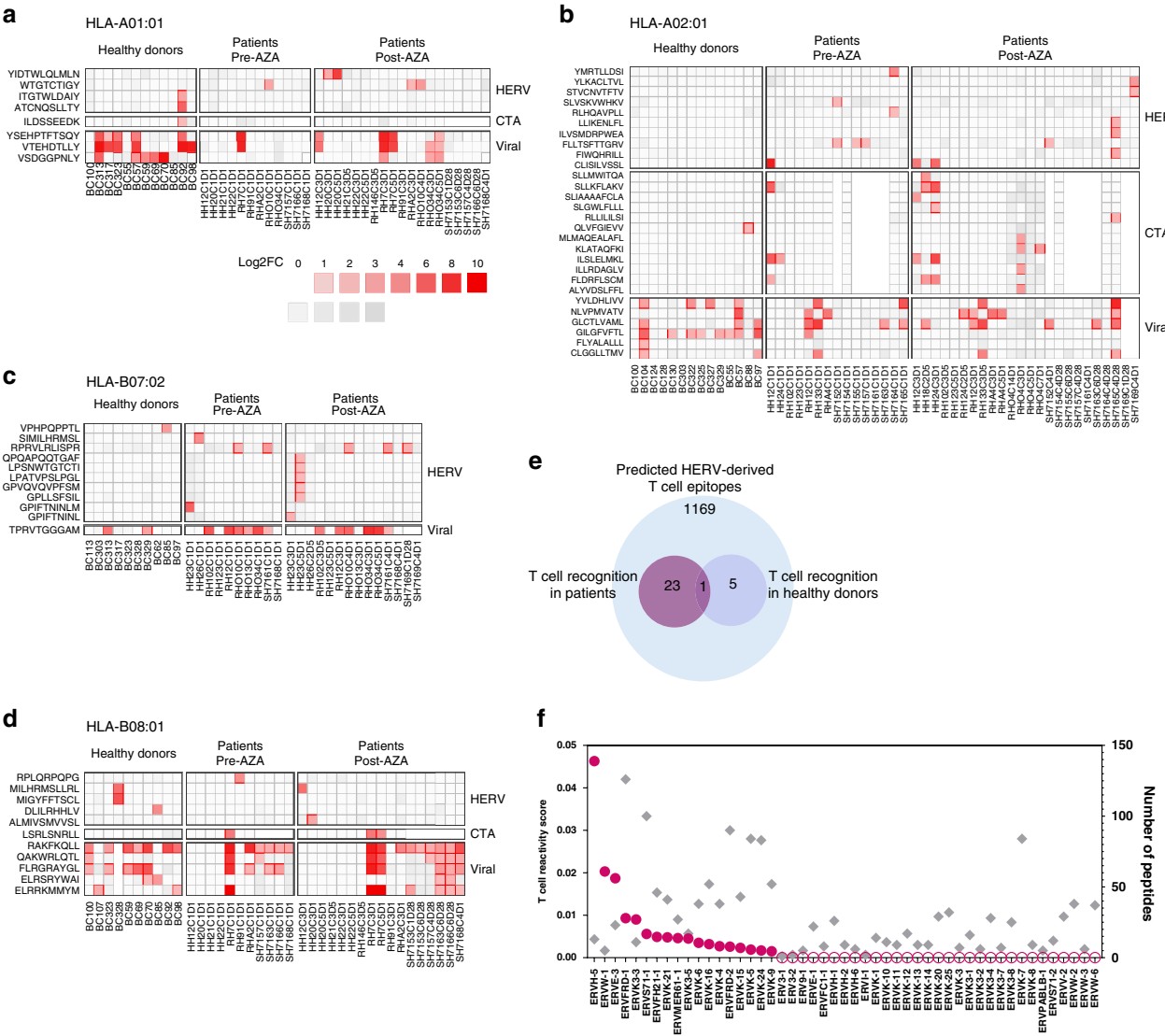

**Fig. 2 T cell reactivity to HERV-derived peptides in myeloid malignancies.** T cells reactive to HERV-derived peptides, CTAs, and viral antigens were identified from peripheral blood (Danish patients and healthy donors (HD)) and bone marrow (Australian patients) samples using a DNA barcode-based pMHC multimer analysis. **a–d** Identified T cell responses are shown across the four tested HLAs for healthy donors and patient samples pre- and post-AZA therapy. Vertical axis labels the sample IDs and the horizontal axis shows the peptide sequences (single letter amino acid codes) split into three categories of antigens (HERV, CTA, and viral). T cell responses are shown based on barcode enrichment (Log2FC) in the sorted population compared to the complete pMHC library, and red scale determines significant enrichment (FDR < 0.1%) and gray scale if no significant enrichment was found. Peptides identified to have a T cell response in at least one of the analyzed samples are included; data for all the tested peptides are shown in Supplementary Fig. 2. The white color indicates peptides not tested in the specific samples. Patient samples with the prefix "RH" and "HH" derive from the Danish patients and patient samples with the prefix "SH" derive from the Australian patients. **e** Venn diagram summarizing numbers of immunogenic HERV-derived T cell epitopes identified in patients and healthy donors. **f** T cell reactivity score (pink dots, plotted in descending order) and peptide library size (gray diamonds) of individual HERVs analyzed for T cell recognition in the patients. Pink hollow dots represent HERVs with no T cell reactivity. T cell reactivity score is calculated as the sum of all the T cell reactive peptides out of the total peptide library of a given HERV tested across the patient samples. Source data are provided as Source data file.

patient samples showed a higher HERV T cell response level than healthy controls (determined as log fold change in the proportion of HERV-peptides recognized; posterior probability of fold change >0 was 96% for pre-AZA and 100% for post-AZA; Fig. 3c). The model further indicates that treatment with AZA may further increase the HERV T cell response, although the statistical certainty is less pronounced (posterior probability that post-AZA response is higher was 82.8%; Fig. 3c). Consistently, we found increased T cell responses following treatment initiation in eight patient samples (Supplementary Fig. 3b). However, post-treatment samples also showed loss of HERV-reactive T cells that were identified in pre-treatment samples (Supplementary Fig. 3b).

In contrast to the increased HERV-specific T cell responses in patients, we observed a significant decrease in CD8+ T cells reactive to the most common viral antigens investigated in our study (Fig. 2 and Supplementary Data 4) as compared to healthy donors (Fig. 3d). The presence of T cells reactive to these common viral antigens could be indicative of the individual's overall immune status; thus, a decrease in such virus-reactive T cell populations in patients suggests an immunosuppressive

**Table 1 HERV-derived T cell epitopes identified based on CD8+ T cell recognition in patients and healthy donors.**

| HERV gene ID | Peptide | HLA | Patients or HDs |
|---|---|---|---|
| ERVFRD-1 | WTGTCTIGY | HLA-A01:01 | Patient |
| | STVCNVTFTV | HLA-A02:01 | Patient |
| | LPSNWTGTCTI | HLA-B07:02 | Patient |
| | GPIFTNINLM | HLA-B07:02 | Patient |
| | GPIFTNINL | HLA-B07:02 | Patient |
| ERVFRD-2 | GPVQVQVPFSM | HLA-B07:02 | Patient |
| | RPLQRPQPG | HLA-B08:01 | Patient |
| ERVE-3 | YIDTWLQLMLN | HLA-A01:01 | Patient |
| | SLVSKVWHKV | HLA-A02:01 | Patient |
| ERVFH21-1 | SIMILHRMSL | HLA-B07:02 | Patient |
| | MILHRMSLLRL | HLA-B08:01 | Patient and healthy donor |
| ERVK-4, ERVK-9, ERVK-15, ERVK-16 | YMRTLLDSI | HLA-A02:01 | Patient |
| ERVK-16 | FIWQHRILL | HLA-A02:01 | Patient |
| ERVS71-1 | ATCNQSLLTY | HLA-A01:01 | Healthy donor |
| | RLHQAVPLL | HLA-A02:01 | Patient |
| | FLLTSFTTGRV | HLA-A02:01 | Patient |
| ERVH-5 | RPRVLRLISPR | HLA-B07:02 | Patient |
| ERVK-7, ERVK-6, ERVK-21 | ITGTWLDAIY | HLA-A01:01 | Healthy donor |
| ERVK-6 | GPLLSFSIL | HLA-B07:02 | Patient |
| ERVK3-3 | LLIKENLFL | HLA-A02:01 | Patient |
| ERVK3-5 | YLKACLTVL | HLA-A02:01 | Patient |
| ERVK-5 | QPQAPQQTGAF | HLA-B07:02 | Patient |
| ERVK-21 | ALMIVSMVVSL | HLA-B08:01 | Patient |
| ERVK-24 | ILVSMDRPWEA | HLA-A02:01 | Patient |
| ERVMER61-1 | LPATVPSLPGL | HLA-B07:02 | Patient |
| ERVW-1 | CLISILVSSL | HLA-A02:01 | Patient |
| | MIGYFFTSCL | HLA-B08:01 | Healthy donor |
| ERVK-12 | VPHPQPPTL | HLA-B07:02 | Healthy donor |
| ERVK-14, ERVK-11, ERVK-13, ERVK-3, ERVK-24, ERVK-25, ERVK-8 | DLILRHHLV | HLA-B08:01 | Healthy donor |

cancer-associated state. This has previously been reported in higher-risk MDS[39] and other cancers[40]. The reduced reactivity toward viral antigens was observed both as a reduction of the proportion of individuals with such T cells in the different cohorts (Fig. 3d) and in the total number of T cell responses against viral antigens present in each individual (Fig. 3e). Our regression model with HLA correction clearly shows a negative fold change in viral antigen T cell reactivity in the patient groups compared to healthy donors (posterior probability of decrease 100% for pre-AZA, and 94% for post-AZA) (Fig. 3f). Notably, post-AZA treatment samples showed the significant improved total number of viral reactive T cell responses as compared to pre-AZA samples (Fig. 3e), which was further supported by the HLA-corrected analysis (95.8% probability) (Fig. 3f). Overall, this suggests a possible AZA-driven improvement of the immune status.

The patient group displayed reduced T cell reactivity toward viral antigens, but simultaneously had an increased T cell reactivity against HERV-derived peptides compared to the healthy donors. This demonstrates that the HERV response is not merely driven by a general increase in T cell reactivity, but in fact is present despite an overall reduced responsiveness of the immune system. Using the viral response to normalize the HERV response (essentially using it as an internal control for overall T cell reactivity) makes the increase in HERV-reactive CD8+ T cells in patients even clearer (for both pre- and post-AZA samples there is a 100% posterior probability for an increase compared to healthy controls; Fig. 3g). The analysis of the normalized response also shows that, to the extent that AZA treatment leads to an increase in the HERV response in cancer patients, this can be explained as a result of the overall increase in T cell responses in these patients (the normalized log fold change between pre- and post-AZA samples is estimated to be very close to 0).

Similar to HERVs, we also observed T cells reactive to CTAs in both patients, pre- and post-AZA, and healthy donors (not significant, Supplementary Fig. 3c). However, these tended to be less frequent and a comprehensive, comparative evaluation would require analysis of a larger cohort.

In summary, we observed a significant enrichment of HERV-reactive T cells in patients with myeloid malignancies despite a general immunosuppressive state as observed by decrease in viral reactive T cells in the patients.

**HERV-specific T cells recognize myeloid cancer cells and show peptide dependent activation.** We validated the presence of HERV-specific CD8+ T cells (HERV-K; HLAB*08:01-DLILRHHLV) in a healthy donor sample using an independent flow cytometry analysis based on fluorochrome-labeled pMHC multimers (Fig. 4a). Since the T cell response identified in this healthy donor was of low frequency and intensity, we further confirmed the presence of these T cells by expanding them ex vivo using a cocktail of pMHCs and cytokines. This strategy resulted in a seven-fold expansion over the initial frequency (Fig. 4a, bottom row).

Using similar approach for functional analyses, we expanded a HERV-specific T cell population identified in one of the patient samples. The expanded T cells from patient SH7152 (CMML), that recognized HLA-A02:01-restricted peptides SLVSKVWHKV (ERVE-3) and FLLTSFTTGRV (ERVS71-1) (Fig. 2b and Table 1), showed increased cytokine production (IFN-γ and TNF-α) and degranulation (CD107a) upon incubation with HLA-matching leukemia cell line THP-1, but not when THP-1 cells were blocked with anti-HLA-A antibodies (Fig. 4b). THP-1 is a human monocytic cell line known to express HERVs[41] and is derived from acute monocytic leukemia. Thus, THP-1-driven activation

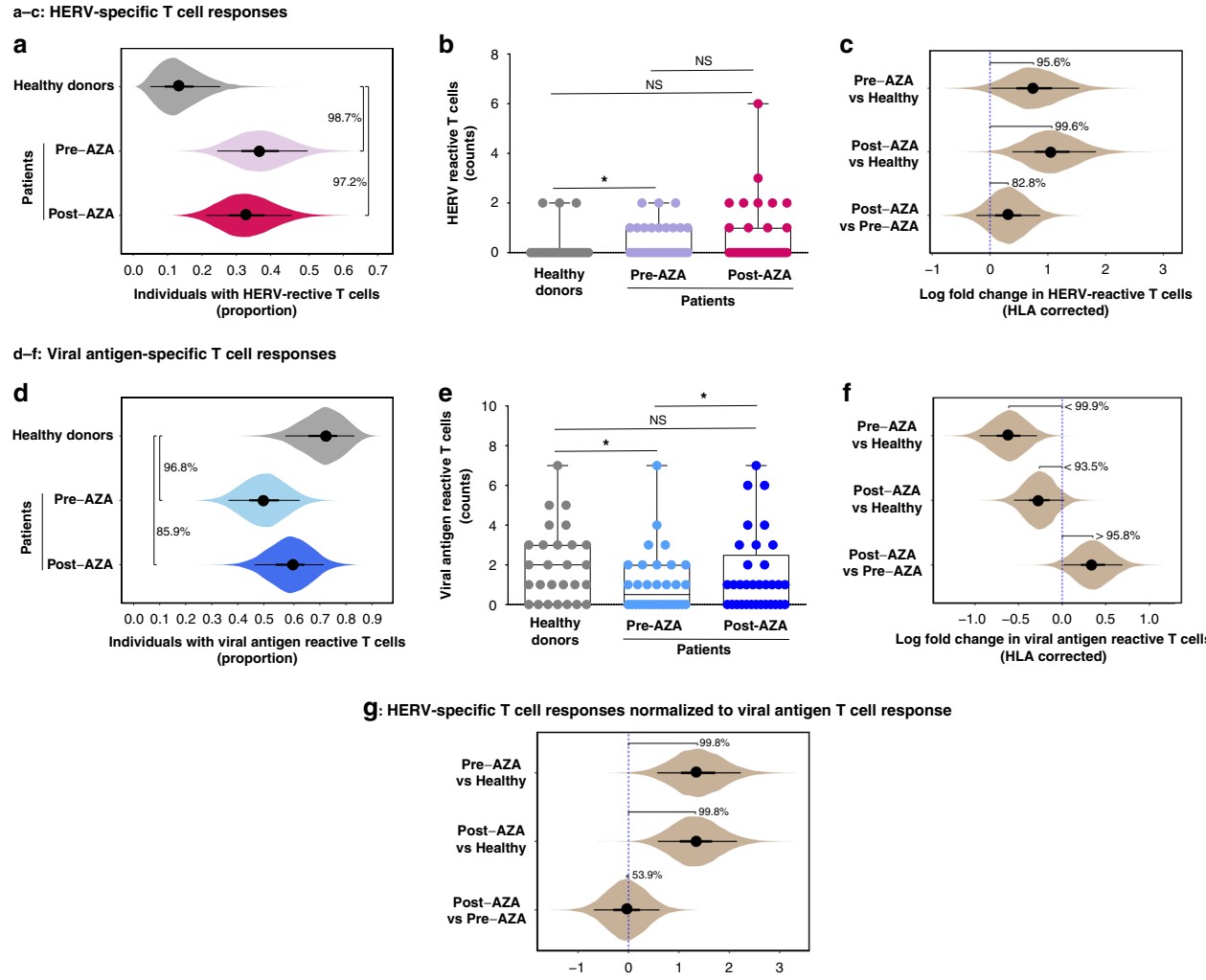

To investigate the relationship between clinical outcome and T cell response to viral or HERV peptides, we used logistic regression. Specifically, the aim with this analysis was to predict clinical outcome (treatment responder or non-responder), based on the following predictors: (1) whether the patient had any T cell response to HERV peptides, (2) whether the patient had any T cell response to viral peptides, (3) an interaction term indicating if a patient had response to both HERV and viral peptides, and finally (4) indicator variables for the presence of each of the 4 HLA alleles. None of these variables had a statistically significant impact on predicting the outcome (the credible intervals for their regression coefficients all include zero, Fig. 4d and Supplementary Fig. 4c, d). This is possibly partly due to the small size of the data set and the resulting low statistical power, and the results indicate that there may be a positive relationship with response to viral peptides and also with the interaction term, but certainty is marginal and require further investigation on a larger cohort.

**Transcription of HERV elements is associated with malignancy.** We next investigated if the induced HERV-specific T cell responses that we observed in patients compared to healthy donors correlated with a higher expression of HERVs. We analyzed the expression of the 49 HERVs used for T cell evaluation at RNA transcript level in bone marrow (BM) CD34+ hematopoietic stem and progenitor cells (HSPCs) from the Australian cohort of patients ($n = 18$)[42]. Of these 18 patients, 12 patients

of T cells specific to HERVs derived from a CMML patient validates the functionality of HERV-specific T cells in the context of myeloid malignancies. To further confirm the peptide-specific activation of HERV-specific T cells, we evaluated the cytokine release following co-culture with target cells pulsed with either HERV-peptides (SLVSKVWHKV and FLLTSFTTGRV) or an irrelevant peptide (HIV epitope ILKEPVHGV). Cytokine release was observed only when target cells were loaded with HERV-derived peptides, suggesting a HERV-peptide-specific functional activation of these T cells (Fig. 4c).

In order to understand whether HERV sequences are, in fact, translated into protein and whether peptides could be presented by leukemia cells in the context of HLA class I, we re-analyzed a published dataset (PMID: 31291378) of HLA-I associated ligandomes of primary AML tumor material and the OCI-AML cell line. We identified peptide sequence TTQEAEKLLER, which originates from the ERK3-1 transcript, in two patients (IDs 005686 and 005685). Peptide TEQGPTGVTM, also originating from ERK3-1, was identified in OCI-AML cells. Both of these peptides were part of our HERV library selected for T cell analysis (Supplementary Data 2). These data further demonstrate that HERV transcripts are expressed and enter the HLA-I presentation pathway in AML. Altogether, we demonstrate that HERV-derived peptides are presented in myeloid malignancies, which can raise HERV-specific CD8+ T cell responses that are functionally active and show HLA and peptide dependent activation.

**Fig. 3 Patients with myeloid malignancies have increased levels of T cell recognition to HERV-derived peptides.** CD8[+] T cell responses, identified using DNA-barcoded-pMHC multimers (Fig. 2), grouped for individual categories of HERV- and viral-antigen libraries across healthy donors and patient samples before and after AZA treatment. **a** The proportion of individuals within the respective groups with detectable HERV-specific T cell responses (in **a**, **c**, **d**, **f**, and **g** uncertainty about estimates is indicated by showing the posterior probability distribution in the form of "eye plots": dot and bars indicate the posterior median, the 50% credible interval (CI), and the 90% CI values). Median and 90% CI values are: healthy donors 0.13 [0.05, 0.25], pre-AZA 0.37 [0.25, 0.50], post-AZA 0.33 [0.20, 0.46]. Posterior probability that proportions increased: 99% (pre-AZA > healthy donors), 97% (post-AZA > healthy donors), and 36% (post-AZA > pre-AZA). **b** The number of HERV peptides recognized by CD8[+] T cell populations in individual healthy donors and patients (pre- and post-AZA). $P$-values for hypothesis tests comparing the number in pairs of groups: $p = 0.02$ (healthy donor vs pre-AZA, Mann–Whitney–Wilcoxon test, one-tailed), $p = 0.07$ (healthy donor vs post-AZA, Mann–Whitney–Wilcoxon test, one-tailed), and $p = 0.60$ (pre-AZA vs post-AZA, Wilcoxon Signed-Rank test, one-tailed). Box plots showing the median, the lower and upper quartiles, and the whiskers as minimum and maximum values. Healthy donors, $n = 27$; pre-AZA, $n = 33$; post-AZA, $n = 34$ (source data are provided as Source data file). **c** Log fold change in proportion of HERV peptides recognized by CD8[+] T cells. The proportion of recognized HERV peptides, and the log fold change in these proportions between pairs of cohorts, was estimated using a regression model that also corrected for the HLA alleles present in each individual sample. Pre-AZA vs healthy donors 0.75 [0.02, 1.5], post-AZA vs healthy donors 1.1 [0.39, 1.8], and post-AZA vs pre-AZA 0.32 [−0.24, 0.87]. Posterior probability that log fold change >0: 96% (pre-AZA vs healthy donors), 100% (post-AZA vs healthy donors), and 83% (post-AZA vs pre-AZA). **d** The proportion of individuals within the respective groups with detectable T cell responses to viral antigens. Healthy donors 0.73 [0.58, 0.85], pre-AZA 0.50 [0.36, 0.64], post--AZA 0.60 [0.47, 0.73]. Posterior probability that proportions in pairs of groups are different: 97% (pre-AZA < healthy donors), 86% (post-AZA < healthy donors), and 80% (post-AZA > pre-AZA). **e** Number of viral antigens recognized by CD8[+] T cell populations in individual patients (pre- and post-AZA treatment) and healthy donors. $P$-values for hypothesis tests comparing the number in pairs of groups: $p = 0.01$ (healthy donor vs pre-AZA, Mann–Whitney–Wilcoxon test, one-tailed), $p = 0.097$ (healthy donor vs post-AZA, Mann–Whitney–Wilcoxon test, one-tailed), and $p = 0.05$ (pre-AZA vs post-AZA, Wilcoxon Signed-Rank test, one-tailed). Box plots showing the median, the lower and upper quartiles, and the whiskers as minimum and maximum values. Healthy donors, $n = 27$; pre-AZA, $n = 32$; post-AZA, $n = 33$ (source data are provided as Source data file). **f** Log fold change in proportion of viral peptides recognized by CD8[+] T cells estimated using the regression model and corrected for individual HLA alleles. Pre-AZA vs healthy donors −0.61 [−0.95, −0.29], post-AZA vs healthy donors −0.26 [−0.55, 0.023], and post-AZA vs pre-AZA 0.35 [0.017, 0.69]. Posterior probability that log fold change >0: 100% (pre-AZA < healthy donors), 94% (post-AZA < healthy donors), and 96% (post-AZA > pre-AZA). **g** Log fold change in proportion of HERV peptides recognized by CD8[+] T cells corrected for HLA alleles and normalized to viral antigen responses; pre-AZA vs healthy donors 1.4 [0.57, 2.23], post-AZA vs healthy donors 1.3 [0.58, 2.2], and post-AZA vs pre-AZA −0.04 [−0.69, 0.61]. Posterior probability that log fold change >0: 100% (pre-AZA vs healthy donors), 100% (post-AZA vs healthy donors), and 54% (post-AZA vs pre-AZA).

were also included in the T cell analyses. HERV expression measured as transcripts per million of total reads (TPM) identified for each HERV in the patients were compared with the transcripts from healthy donor BM HSPCs ($n = 14$; reported previously[43–45]) (Supplementary Data 5). A strong malignancy-associated increase was found in the majority of the HERVs selected for the T cell analyses (Fig. 5a), which aligns with our observation of a higher HERV-specific T cell recognition in patients versus healthy donors. This enhanced expression of the analyzed HERV transcripts was found in samples from both before and after AZA treatment. An additional, but low, increase was further observed after AZA treatment when comparing the level of each individual HERV pre- and post-therapy in each individual patient (Fig. 5a), but no overall increase was observed when assessed as the sum of HERVs across the evaluated individuals (Supplementary Fig. 5a, b). However, a patient-specific increase in HERV expression post-AZA treatment was observed in few of the analyzed samples (Supplementary Fig. 5c).

Importantly, we observed that the level of T cell recognition is correlated to the level of expression of the given HERVs recognized ($r = 0.47$, $p = 0.018$; Fig. 5b). This indicates that the amount of antigen does play a role for induction of T cell recognition. Furthermore, at a gross level across all the HERVs evaluated for T cell recognition, we found a tendency toward transcript levels being higher for those HERVs recognized by T cells compared to those not recognized (median 2.06 vs 1.19; Fig. 5c), indicating that indeed, higher HERV expression may lead to enhanced levels of HERV T cell recognition. However, there was a large variation in HERV expression and hence no significant distinction between the expression levels for T cell 'immunogenic' versus 'non-immunogenic' HERVs.

Comparison of expression levels in healthy donors to untreated patient samples showed a substantial induction of almost all the 49 HERV transcripts in the patients, suggesting a strong disease-related expression pattern (Fig. 5d). Further, we evaluated if AZA treatment influenced the expression levels of these 49 HERVs in addition to the induction seemingly caused by the disease. We compared RNA-seq data from BM HSPCs collected before and after AZA treatment (at cycle 6 or at the end of the trial) and observed that AZA-treatment led to enhanced expression of some HERVs, but repression of others. Furthermore, the effect was very heterogeneous among the patients evaluated. Hence, no overall change was observed related to AZA treatment (Fig. 5d, e and Supplementary Fig. 6a).

Since a number of previous studies have shown upregulation of silent HERV transcripts upon short-term in vitro treatment with HMAs[16,17], we further examined if any additional changes in the expression of our selected HERVs could be observed after only short-term in vivo treatment (pre-treatment compared with 1 and 4 weeks after treatment initiation). However, no overall change in the HERV expression was present at these earlier time-points (Supplementary Fig. 6b). For some individual patients, we did observe an increase in HERV expression over time with AZA treatment, but in such cases, the later time point displayed the most prominent upregulation (Supplementary Fig. 6c). This suggests a minimal impact of AZA treatment, both short and long term, on the expression of these particular 49 HERVs beyond the disease-driven upregulation.

Additionally, as we observed several CTA-reactive T cells in this study (Supplementary Fig. 3a), we investigated the expression of a large number of known CTAs (CTDatabase) similarly to the HERV analysis. We found malignancy-associated upregulation of some CTA genes (SSX4, SSX4B, PRAME, ROPN1, etc.) in the patient cohort compared to healthy donors (Supplementary Fig. 7).

Altogether, presence of HERV-specific T cells strongly correlated with the expression of HERVs, which, for the 49 HERVs analyzed in this study, was upregulated in patients before exposure to DNA-demethylating therapy.

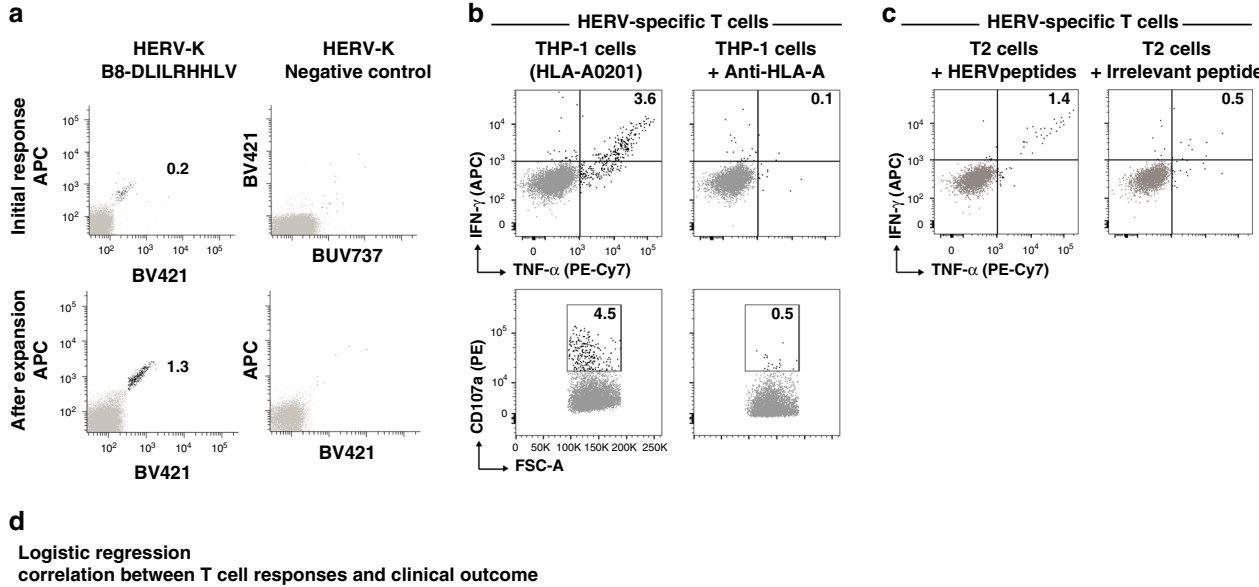

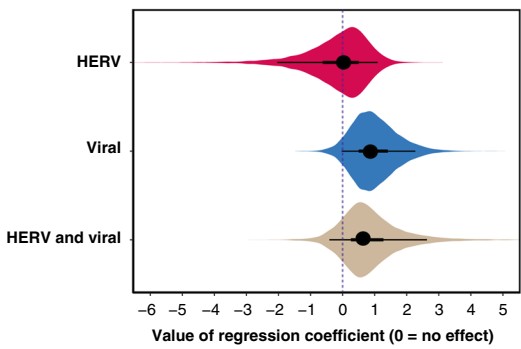

**Fig. 4 HERV-reactive T cell functionality and their relation to clinical outcome. a** Top panel, Flow cytometry plots of T cells detected in healthy donor PBMCs (BC85) reactive to a HERV-derived peptide (identified using DNA-barcoded multimer analysis, shown in panel d). An HLA-B*08:01-restricted HERV-derived peptide not identified to be reactive with T cells in the DNA-barcode multimer analysis was used as a negative control. Bottom panel, HERV-specific T cells from the same donor were expanded ex vivo using pMHCs and cytokines. T cells were identified using pMHC tetramers in a combinatorial color encoding strategy defining a unique two-color combination for identified T cells. Black dots represent dual-color tetramer-positive antigen-specific CD8+ T cells and gray dots are tetramer-negative or single-color positive CD8+ T cells. Numbers on plots indicate frequency (%) of antigen-specific CD8+ T cells. **b** Functional analysis of HERV-specific T cells expanded from patient SH7152 (identified using DNA-barcoded multimer analysis, shown in **b**) using pMHCs (SLVSKVWHKV and FLLTSFTTGRV restricted to HLA-A*02:01) incubated with HLA-A*02:01-restricted THP-1 cells. THP-1 cells blocked with anti-HLA-A antibodies are used as controls. **c** Functional analysis of same HERV-specific T cells in the presence of T2 cells loaded with HERV-peptides SVSKVWHKV and FLLTSFTTGRV or with an irrelevant peptide (A*02:01-restricted HIV pol epitope ILKEPVHGV). Numbers on the plots (B and C) show the frequency of CD8+ T cells positive for intracellular cytokine IFN-γ and TNF-α and degranulation marker CD107a. **d** Logistic regression exploring the relationship between clinical outcome and T cell response to viral or HERV peptides. The plot shows the posterior distribution of regression coefficients (eye plot with indication of posterior mean, 50% CI, and 90% CI) for the following predictors: HERV; the effect of any response to HERV peptides. Viral; the effect of any response to viral peptides. HERV and viral; interaction term indicating the effect of having T cell responses to both HERV and viral peptides. Note that the credible intervals for all regression coefficients include 0, indicating no statistically significant effect of the explored predictors, although there may be a positive effect of Viral and HERV and viral.

## Discussion

In this study, we investigated the presence of T cells specific to HERV-derived peptides in the myeloid hematological malignancies MDS, CMML, and AML. We evaluated 66 HERV loci for T cell recognition. 1169 peptides from 49 of these 66 HERVs were predicted to bind the selected four HLA molecules (HLA-A*01:01, -A*02:01, -B*07:02, and -B*08:01). We determined the T cell recognition of these 1169 potential T cell epitopes in both patients and healthy controls, and demonstrated the presence of T cell populations recognizing 29 novel HERV-derived epitopes of which 23 were unique to the patient cohort. Among the evaluated HERVs, HERVH-5, HERVW-1, HERVE-3, and several members of HERVK family were primarily recognized by T cells with a high T cell reactivity score, which may be a consequence of

the high expression levels of these HERVs. Our data suggest a disease-specific enrichment of CD8+ T cells recognizing peptides of HERV origin which is consistent with the observed change in HERV expression patterns in patients compared to healthy donors. This was evaluated by combining T cell recognition identified in both PBMCs and BMMCs of the two patient cohorts, as essentially both sampling tissues are part of similar circulating malignant compartment and we did not observe any tissue-specific difference in the level of identified T cells. A sample specific comparison between patients and healthy controls would require a larger cohort of patient samples.

Previously, CD8+ T cells targeting HERV-derived peptides have been reported in a few different cancers[46]. For instance, Takahashi et al. reported a HERV-E-derived 10mer antigen as a T

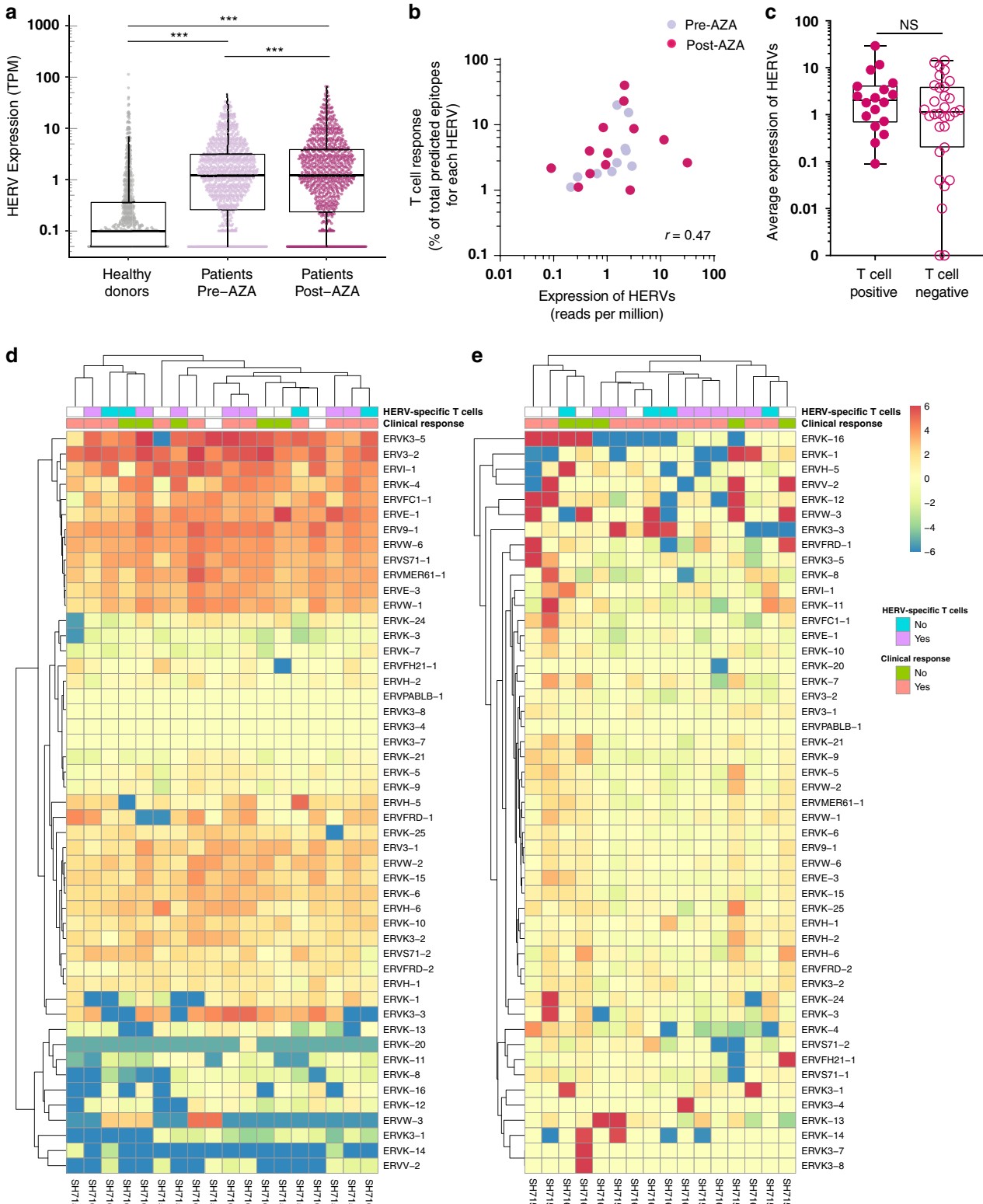

cell target for metastatic renal cell carcinoma demonstrating an example of anti-tumor activity of HERV-derived T cells[27]. The study presented here, is the first comprehensive analysis of T cell recognition of a large library of HERVs, revealing that on a broad scale these are potential targets for T cell recognition in cancer. For hematological malignancies, to our knowledge, no previous study has demonstrated CD8[+] T cells specific to HERV-derived peptides. Interestingly, one of the HERV-derived peptides

included in our library was earlier found to be active in another tumor context (FLQFKTWWI; HERV-K-derived and HLA-A*02:01- and HLA-B*08:01-restricted, for which CD8[+] T cell reactivity was detected in a study including patients with semi-noma[29]). However, none of the 29 HERV-derived peptides identified as targets for T cell recognition in this study has, to our knowledge, been found active in other contexts. Besides this, we could also confirm the expression and HLA-I presentation of two

**Fig. 5 Enrichment of HERV-specific T cells in patients correlates with enhanced expression of HERV elements.** RNA-seq analysis of previously published data of 18 patients from the Australian patient cohort was performed to quantify the expression of the 49 HERVs tested for T cell analysis. **a** Expression of the 49 HERVs in each of the patients (before, $n = 18$, and after, $n = 16$, AZA treatment) compared with healthy donors ($n = 14$). Mann–Whitney–Wilcoxon test, two-tailed, $p < 2.2e-16$ (healthy donors vs pre-AZA), $p < 2.2e-16$ (healthy donors vs post--AZA). Wilcoxon Signed-Rank test, two-tailed, $p = 1.8e-4$ (pre-AZA vs post-AZA). RNA-seq data of healthy donors were obtained from previously published data (Supplementary Table 6). Healthy donors, $n = 588$ (49 HERVs across 12 healthy donors; pre- and post-AZA, $n = 784$ (49 HERVs across 16 patients). **b** Correlation of HERV-specific T cell responses identified in patient samples (shown in Fig. 2) with the expression of their associated HERVs. X-axis shows expression of the HERV transcripts (mean across the 18 patients, before or after treatment). Y-axis shows the proportion of identified T cell epitopes (before and after AZA treatment), i.e., number of T cell epitopes detected out of total predicted epitopes from a given HERV. Spearman correlation between HERV expression and T cell responses, $r = 0.47$ was significantly different from zero ($p = 0.018$) (source data are provided as Source data file). **c** Expression of the 18 HERV loci recognized by T cell populations compared with the expression of the 31 HERV loci not contributing to any T cell response. Expression was quantified as transcripts per million of total reads (TPM) and is given as the mean value for patients, including values both before and after AZA treatment; Mann–Whitney–Wilcoxon test, two-tailed, $p = 0.26$. T cell positive, $n = 18$; T cell negative, $n = 31$ (source data are provided as Source data file). **d** Heatmap of estimated fold change for individual HERVs across the 18 patients (before treatment) compared to mean expression HERV value across all healthy donors (using mean expression values of transcripts for each HERV gene). For comparative representation, the fold change color scale is restricted from $-6$ to $6$, and fold change values outside this limit are shown at the maximum ($>6 = 6$) or minimum scale ($<-6 = -6$). HERV specific T cell occurrence together with information regarding clinical response is annotated in the top bars for each patient. **e** Similar to (**d**) comparing pre- and post-AZA treatment. In boxplots the box shows the 1st quartile (Q1), the median, and the 3rd quartile (Q3), while whiskers extend to 1.5 times the interquartile range (IQR) on either side of the box (or to the minimum and maximum data values if these are less than 1.5 * IQR from Q1 and Q3).

of our predicted HERV-derived peptides in patients with AML and in an AML cell line using mass spectrometry.

HMAs, such as AZA, have been shown to derepress HERV elements in vitro and recently also in vivo, and in turn activate the viral response pathway through the formation and detection of viral dsRNAs, which may increase the immunogenicity of the malignant cells[16–19,47,48]. Additionally, this pathway may enhance the visibility of the tumor cells to the immune system through increased MHC I expression and antigen processing[49]. On the other hand, HMAs may also upregulate immune checkpoints directly or as a compensatory mechanism, which potentially hamper the immune responses[50,51]. Therefore, this HERV-associated antigen reservoir could be of immense interest for immunological targeting of myeloid malignancies as it presents a unique opportunity for T cell-mediated therapeutic intervention when combining HMAs with cancer immunotherapies[13]. In our cohort of 34 patients treated with AZA monotherapy, we observed only a low level of treatment-induced upregulation of the transcripts corresponding to the 66 selected HERVs and a marginal enhancement of HERV-reactive T cells was observed. However, it is possible that our HERV library did not capture the predominantly DNA methylation-regulated HERVs since they were selected based on transcriptional activity without any prior treatment with HMAs. Nonetheless, our data show that the observed T cell recognition correlated with the corresponding HERV RNA expression, and hence T cell responses to DNA methylation-regulated HERVs may still be induced by HMAs. Indeed, a patient tailored treatment strategy with selection of HERVs specifically silenced by DNA methylation and derepressed by HMAs may be successful in combination with other immune stimulatory treatments, such as immune checkpoint blockade.

Recent studies have shown that HERV expression is associated with tumor immune cytolytic activity[12] as well as improved clinical response to immunotherapy in patients with cancers[32,52,53]. Our results indicate that the malignant cells of certain myeloid origins harbor increased levels of certain HERV transcripts and could serve as a potential antigen pool for tumor-targeting immunotherapies. Furthermore, our data suggest that T cell recognition of HERVs may support the immune reconstitution following HMA treatment, potentially by mediating cancer-cell control.

For immune checkpoint blockade, there is a robust association between tumor mutation burden and treatment response[54,55]. The mutations in tumor cells can give rise to neoantigens, which are recognized as non-self-epitopes and thereby enhance the immune reactivity against tumor cells. Arguably, HERV-derived

epitopes, may act in a similar manner and translate to immunogenic (neo)antigens, potentially with a higher degree of similarity to known pathogenic epitopes, which may predict better responses[56]. Moreover, HERV-derived epitopes may serve as a shared pool of antigens expressed across tumors from different patients, as compared to neoantigens that are almost exclusively personal and furthermore subjected to a large degree of intra-tumoral heterogeneity[57]. Therefore, HERVs may serve as a reservoir of targetable antigens in myeloid malignancies that generally carry a low mutational burden[58]. In this study, we found that the investigated HERV loci shared a rather similar regulation at the RNA level across patients before undergoing epigenetic therapy, whereas the expression levels after treatment were more heterogeneous. This may reflect that AZA is incorporated in the DNA of the patients' malignant cells to different degrees, possibly depending on their division pace and on the specific activity of the intracellular enzymes responsible for AZA degradation. Moreover, the activity and efficiency of the intracellular mechanisms responsible for the degradation of HERV RNA molecules could also vary among the patients. Obviously, the relative influence of DNA methylation versus histone modifications on the expression of the specific HERVs may vary among the patients due to a disrupted cancer epigenome and different evolutionary ages of the specific HERVs[19,24]. Indeed, as discussed earlier, these specific HERVs may only to a low degree be regulated by DNA methylation. Lastly, the subcellular and clonal composition of the malignant cells will change unequally in the patients after subjection to AZA treatment, demanding single cell analyses.

Currently, a number of ongoing clinical trials are designed to determine the impact of combining HMAs and immune checkpoint inhibitors, as well as epigenetic therapy and immunotherapy in general, both in hematological malignancies and solid cancers[13,59]. Our data support the need to study HERV-derived T cell recognition in more detail in these treatment combinations to determine the direct impact of T cell recognition of HERV-derived peptides on tumor regression.

## Methods

**Study plan.** The study was designed to investigate the immunogenicity of HERVs in the context of cancers with low mutational burden, with the hypothesis of identifying a novel class of antigens reactive to CD8+ T cells and their potential application in cancer immunotherapy. A comprehensive analysis of HERVs resulted in the prediction of 1,169 potential epitopes that were used to identify CD8+ T cells in patients with hematological malignancies.

Patient samples were selected to represent hematological malignancies with a relatively low mutational burden (MDS, AML, CMML), and represented peripheral

blood (PB) ($n = 22$) and bone marrow (BM) ($n = 12$) samples to compare a source dependent analysis of antigen-reactive CD8[+] T cells. Samples from 27 healthy donors were included as controls. Pre- and post-treatment samples from individual patients treated with HMAs were used to segregate malignancy- and therapy-driven immunogenicity of HERV-derived peptides. A well-established DNA-barcode based multiplex technology was used for CD8[+] T cell detection in the patient and healthy donor samples.

Standard bioinformatics tools were used for RNA-seq analysis of HERVs and CTAs in the patient cohort and compared with RNA-seq data from healthy donors (see RNA-seq data analysis).

**Patients and sample collection**. The PB samples derived from 22 Danish patients with MDS ($n = 13$), CMML ($n = 4$), and AML ($n = 5$), and from 27 healthy donors, all carrying one or more of the four most common HLA alleles in Caucasian populations (HLA-A*01:01, HLA-A*02:01, HLA-B*07:02, and HLA-B*08:01). The patients were selected based on their sample availability before and following AZA treatment and their HLA type. The patient PB samples were collected at Rigshospitalet and Herlev Hospital, Copenhagen University Hospitals, Denmark. All uses of human material have been approved by the committee on health research ethics in the Capital Region of Denmark and the human research ethics committee of the South Eastern Sydney Local Health District, Australia, and informed written consent was obtained in accordance with the Declaration of Helsinki. All patients were diagnosed according to the World Health Organization (WHO) criteria[60], and patients with MDS and CMML were risk stratified using the Revised International Prognostic Scoring System (IPSS-R)[61] and CMML-specific prognostic scoring systems (CPSS)[62], respectively. The clinical treatment responses were evaluated in accordance with the International Working Group (IWG) response criteria[63], as complete remission (CR), marrow complete remission (mCR), partial remission (PR), hematological improvement (HI), stable disease (SD), or progressive disease (PD). Overall response was defined as CR, mCR, PR, and HI and determined as the best response during the treatment course. The patients were treated with 5-azacytidine (AZA; Vidaza, Celgene, NJ) at a dose of 100 mg/m2/day s.c. for five consecutive days on a 4-week cycle according to the Nordic MDS guidelines (http://www.nmdsg.org). PB samples were as a rule collected before AZA administration on the first and fifth day during each individual treatment cycle.

The BM samples derived from 12 patients with MDS ($n = 7$) and CMML ($n = 5$) that were enrolled in a clinical study in New South Wales, Australia[42]. As for the Danish patients, the Australian patients were selected based on their sample availability before and following AZA treatment and their HLA type. The relevant institutional ethics committees approved the study and informed written consent was obtained from the patients. These patients were treated with AZA at a dose of 75 mg/m2/day s.c. for seven consecutive days on a 4-week cycle and BM samples were collected at defined time points.

Peripheral blood mononuclear cells (PBMCs) from the Danish patients were isolated from PB immediately after sampling using Ficoll-Paque PLUS (GE Healthcare) density gradient centrifugation and were cryopreserved thereafter. PBMCs from healthy donors and BM mononuclear cells (BMMCs) from the Australian patients were isolated by density gradient centrifugation using Lymphoprep (StemCell Technologies). CD34[+] cell enrichment for RNA-seq was carried out on the samples from the Australian patients by using CD34[+] magnetic beads and an AutoMACS Pro Machine (Miltenyi Biotec) according to the manufacturer's instructions. The remaining BMMCs were cryopreserved. Live frozen BMMCs from 12 Australian patients could be included in the T cell analysis, whereas samples from the total cohort of Australian patients ($n = 18$) could be included in the RNA-seq analysis.

Healthy donor PBMCs were obtained from the central blood bank, Rigshospitalet, Copenhagen, in an anonymized form. Healthy donor samples for RNA-seq analysis originated from three previously published papers and RNA-seq data were downloaded from the sequence read archive (SRA) using the SRA toolkit (version 2.8.2-1) fastq-dump (Supplementary Data 5).

**HERV peptide library generation**. Mayer et al. provides a list of genbank accession numbers corresponding to 66 HERVs with 1–4 accession numbers per HERV[34]; we interpreted the HERV symbol as the gene name and the individual accession numbers as transcripts that correspond to the same HERV. The list of accession numbers enabled downloading of the nucleotide sequences using a custom python script applying the BioPython Entrez.efetch function. The script used the relevant accession numbers to download the sequences directly from NCBI's Entrez database. This generated a FASTA file in which the gene symbols (identified by Mayer et al.) were added to each individual sequence header. Each entry in the FASTA file was translated from nucleotide to amino acid sequence using the first occurring start-codon until a potential stop codon occurred, applying the BioPython functions Seq.Seq, Alphabet.generic_dna, and dna_sequence.translate. The peptide sequences were chopped into overlapping peptides of 9–11 amino acids, and the binding affinity to HLA-A*01:01, HLA-A*02:01, HLA-B*07:02, and HLA-B*08:01 was predicted using NetMHCpan2.8[35]. Peptides with a percentile rank score ≤2 were selected and included in the final peptide library. This was based on the authors' guidelines of using netMHCpan 2.8 where peptides determined as binders have a percentile rank score ≤2 (weak binders) and ≤0.5 for

strong binders. The percentile rank for a peptide is generated by comparing its score against the score-distribution of a large set of random natural peptides; i.e., a rank score of 1% means that the predicted score of the peptide is equal to the top 1 percentile score of the large set of random natural peptides. The percentile rank score is recommended, compared to the predicted binding affinity, to ensure predictive compatibility across different HLA alleles. The predicted binding affinities and percentile rank scores were annotated for each peptide HLA pair, together with the HERV gene(s) of the peptide origin and the number of occurrences for the individual peptides across different HERVs. The resulting CSV file was sorted according to the predicted percentile rank score (Supplementary Data 2).

**Cancer testis antigens and viral antigens**. A cancer testis antigen (CTA) library was developed based on a literature search and CTAs compiled into a Ctdatabase (http://www.cta.lncc.br)[64]. Our library covered 67 well defined CTAs of 8–11 amino acid long peptides that are reported to be associated with several cancer types (peptide sequence and HLA restriction are described in Supplementary Data 3).

Similarly, a peptide library of 19 known exogenous viral epitopes, representing antigens from cytomegalovirus (CMV), Epstein-Barr virus (EBV), human immunodeficiency virus (HIV), and influenza (Flu) viruses were prepared to analyze cell reactivity against any of these common viral antigens (peptide sequences and HLA restriction; Supplementary Data 4).

**Peptides**. All peptides of HERV-, CTA-, and viral antigens in the library were custom synthesized by Pepscan Presto BV, Lelystad, The Netherlands. Peptide synthesis was done at a 2 μmol scale with UV and mass spec quality control analysis for 5% random peptides. For the HERV peptide library, custom synthesis was achieved for 1169 out of 1173 peptides by the manufacturer. Peptide stocks were prepared in DMSO at a 10 mM concentration. Ultraviolet light sensitive peptide ligands for MHC class I folding were custom synthesized by the Peptide facility at Leiden University Medical Center (LUMC), The Netherlands, using previously described methods[65–67].

**MHC class I monomer production**. MHC class I monomers of HLA-A*01:01, A*02:01, B*07:02, and B*08:01 were produced using methods previously described[68]. Briefly, MHC class I heavy chain and human ß2-microglobulin (hß2m) were expressed in *Escherichia coli* using pET series expression plasmids. Soluble denatured proteins of the heavy chain and hß2m were harvested using inclusion body preparation. The folding of these molecules was initiated in the presence of UV labile HLA specific peptide ligands. Folded MHC molecules were biotinylated using the BirA biotin-protein ligase standard reaction kit (Avidity, LLC- Aurora, Colorado) and MHC class I monomers were purified using size exclusion chromatography (HPLC, Waters Corporation, USA). All MHC class I folded monomers were quality controlled for their concentration, UV degradation, and biotinylation efficiency and stored at −80 °C until further use.

**DNA barcode-dextran library preparation**. DNA barcodes were prepared using methods described in Bentzen et al.[36], wherein each barcode represents a 5′ biotinylated unique DNA sequence obtained by combining different A and B oligos. These unique barcodes were attached to phycoerythrin (PE) and streptavidin-conjugated dextran (Fina BioSolutions, Rockville, MD, USA) by incubating them at 4 °C for 30 min to generate a DNA barcode-dextran library of 1325 unique barcode specificities.

**T cell staining using DNA barcode tagged peptide-MHC multimers**. HERV peptide library specific monomers, restricted to HLA-A*01:01, A*02:01, B*07:02, and B*08:01, were generated by a UV mediated peptide exchange process[65,66,69,70]. These peptide-specific monomers were then attached to their corresponding DNA barcode dextrans by incubating at 4 °C for 30 min, thus providing a DNA barcode-labeled dextran for each peptide-MHC (pMHC multimer) specifically to detect the respective T cell population. The same process was followed for the CTA- and viral antigen libraries. The T cell staining process used has previously been described[36]. Briefly, pooled pMHC multimers (HLA matching HERV and all the CTA- and viral-specific pMHC dextrans) were incubated with $2–7 \times 10^6$ PBMCs or BMMCs (thawed and washed twice in RPMI + 10% FCS, and washed once in barcode cytometry buffer) for 15 min at 37 °C at a final volume of 80 μL. Cells were then mixed with 20 μL of antibody staining mix containing CD8-PerCP (Invitrogen MHCD0831) or CD8-BV510 (BD 563919) (final dilution 1/50), dump channel antibodies: CD4-FITC (BD 345768) (final dilution 1/80), CD14-FITC (BD 345784) (final dilution 1/32), CD19-FITC (BD 345776) (final dilution 1/16), CD40-FITC (Serotech MCA1590F) (final dilution 1/40), CD16-FITC (BD 335035) (final dilution 1/64), and a dead cell marker (LIVE/DEAD Fixable Near-IR; Invitrogen L10119) (final dilution 1/1000), and incubated at 4 °C for 30 min. Cells were washed twice with barcode cytometry buffer and fixed in 1% PFA.

For confirmation analysis of pMHC specific T cells, 1.5 μL pMHC multimers (of individual specificity) were incubated with $2 \times 10^6$ PBMNCs or expanded T cells and stained using methods described above.

**Identification of T cell reactive peptide-MHC specificities**. Cells fixed after staining with pMHC-multimers were acquired on a FACSAria flow cytometer instrument (AriaFusion, Becton Dickinson). Cells were gated for lymphocytes, singlets, live, and CD8 positives by the FACSDiva acquisition program (Becton Dickinson), and all the PE positive (multimer binding) cells of $CD8^+$ gate were sorted into pre-saturated tubes (2% BSA, 100 µl barcode cytometry buffer) (Supplementary Fig. 8a). Sorted cells belonging to each sample were then subjected to PCR amplification of its associated DNA barcode(s). Cells were centrifuged for 10 min at $5000 \times g$ and the supernatant was discarded with minimal residual volume. The remaining pellet was used as the PCR template for each of the sorted samples and amplified using Taq PCR Master Mix Kit (Qiagen, 201443) and sample specific forward primer (serving as sample identifier) A-key[36]. PCR-amplified DNA barcodes were purified using the QIAquick PCR Purification kit (Qiagen, 28104) and sequenced at Sequetech (USA) or GeneDx (USA) using Ion Torrent PGM 314 or 316 chip (Life Technologies).

**Processing of sequencing data from DNA barcodes**. To process the sequencing data and automatically identify the barcode sequences, we designed a specific software package, 'Barracoda' (https://services.healthtech.dtu.dk/service.php?Barracoda-1.8). This software tool identifies the barcodes used in a given experiment, assigns sample ID and pMHC specificity to each barcode, and calculates the total number of reads and clonally reduced reads for each pMHC-associated DNA barcode. Furthermore, it includes a statistical processing of the data. Details are given in Bentzen et al.[36] The analysis of barcode enrichment was based on methods designed for the analysis of RNA-seq data and was implemented in the R package edgeR. Fold changes in read counts mapped to a given sample relative to mean read counts mapped to triplicate baseline samples were estimated using normalization factors determined by the trimmed mean of M-values. P-values were calculated by comparing each experiment individually to the mean baseline sample reads using a negative binomial distribution with a fixed dispersion parameter set to 0.1[36]. False-discovery rates (FDRs) were estimated using the Benjamini–Hochberg method. Specific barcodes with an FDR < 0.1% were defined as significant. At least 1/1,000 reads associated with a given DNA barcode relative to the total number of DNA barcode reads in that given sample was set as threshold to avoid false-positive detection of T cell populations due to low number of reads in the baseline samples.

**T cell functional analysis**. Patient samples (bone marrow derived) positive for HERV-reactive T cells were expanded using pMHC-scaffolds conjugated with corresponding HERV-derived peptide and cytokines (IL-2 and IL-21) for two weeks in X-vivo media (Lonza, BE02-060Q) supplemented with 5% human serum (Gibco, 1027-106). Expanded T cells were co-cultured with THP-1 cells (with or without anti-HLA-A (Nordic BioSite, ACB-5EBF45) antibodies; final dilution 1/10) or T2 cells (loaded with 10 µM HERV-derived peptide or irrelevant peptide) and incubated for 4 h at 37 °C in the presence of protein transport inhibitor (GolgiPlug; BD Biosciences, 555029; final dilution 1/1000). Functional activation of T cells was measured using anti-CD107a (BD Bioscience, 555801; final dilution 1/40), and intracellular cytokines IFN-γ (BD Bioscience, 341117; final dilution 1/20) and TNF-α (Biolegend, 502930; final dilution 1/20). Cells incubated with Leukocyte Activation Cocktail (BD Biosciences, 550583; final dilution 1/500) were used as a positive control (Supplementary Fig. 8b).

**RNA-seq data analysis**. Samples from 18 Australian patients were used to generate paired-end RNA-seq data (see ref. [42] for RNA-seq procedure). The raw sequencing reads were quality trimmed with an average quality read score of 20, standard adaptors were cut at the first 5 bases of the read due to an unknown sequence skewing the base pair balance. This was done with the wrapper tool trim galore (version 0.4.0), combining FastQC (version 0.11.2) and cutadapt (version 1.9.1). The transcripts downloaded for the library generation from the NCBI Entrez database were added to the ENSEMBL GRCh38 version 85 cDNA transcript reference, to avoid aligning reads of standard genes to HERV-specific genes. The quality checked reads were given as inputs to Kallisto (version 0.42.1) running 500 bootstraps, and following combination of the output files, the average expression value in transcript per million reads (TPM) was used as the output expression value.

All expression values below 0.05 TMP were set to 0.05 TMP to avoid skewing the scale of the wanted visualized area. In the same manner, transcripts with very high expression were set to a defined value, which differ from the plots and is defined in the legend text. All log fold changes were calculated with log2(x)−log2(y). For comparison of healthy donors and samples from patients before treatment, the mean values across all healthy donors were used. For comparison of before and after treatment in patients, the calculated log fold changes were derived from paired samples and only patients with data from both before and after treatment were included. Heatmaps were created using pheatmap (version 1.0.12) in R.

**Statistical analysis**. All statistical analyses were performed using R version 3.6.1[71]. Null hypothesis significance testing for comparisons of paired data were performed using the Wilcoxon signed-rank test, while comparisons of unpaired data were done using the Mann–Whitney-Wilcoxon test, both from the "coin" R-package, which allows computation of exact p-values also in the presence of ties[72]. Bayesian

statistical analyses were performed using the software RStan, version 2.19.3[73,74]. Preparation of data as well as post-processing and plotting of results was done using the "tidyverse" (version 1.3.0) and "tidybayes" (version 2.0.3) R-packages[75,76].

For all Bayesian models (see descriptions below) we ran 3 independent MCMC chains for 10.000 iterations each, with 5.000 iterations warm-up, resulting in 15.000 post-warmup samples for each parameter. Convergence of MCMC runs was monitored by comparing the posteriors from the three independent chains, and it was checked that the potential scale reduction factor[71] ("R-hat") was close to 1 at the end of the run for all model parameters, and that effective sample sizes for all parameters were large (typical values > 5000). Different priors were checked and found to not influence results substantially (Supplementary Fig. 9).

*Bayesian model of proportion of individuals showing an immune response to either HERV or viral antigens (Fig. 3a, d).* It was assumed that each of the three investigated groups (healthy donors, patients pre-AZA, and patients post-AZA) have a typical population proportion of responders. It was further assumed that the observed number of individuals in the samples follow binomial distributions with the sample N and population proportion as parameters. Non-informative, beta(1,1) priors were used for the population proportions. Based on the joint posterior distribution for the three population proportions we could compute the posterior probability for all contrasts (e.g., the posterior probability that the proportion of HERV responders is larger among pre-AZA patients than among healthy donors).

*Bayesian model of the proportion of viral antigens recognized (Fig. 3f, g).* We used essentially the same model as for proportion of individuals, but with proportion now being of viral (not HERV) peptides recognized in a class, instead of proportion of people responding in a class.

*Bayesian regression model of number of HERV peptides recognized by T cell populations, corrected for effect of HLA allele present (Fig. 3c, g).* In this analysis the observed data were the number of peptides recognized by T cells for each individual, for the up to 4 different possible HLA-alleles. We assume that the number of positive peptides is drawn from a binomial distribution where the proportion depends both on the investigated allele and on the class to which the person belongs (healthy donor, patient pre-AZA, or patient post-AZA). Specifically, the population proportion of recognized peptides is assumed to depend on HLA-allele and person-class in the form of a logistic regression model:

$$p_i = \text{logistic}\left(\beta_0 + \beta_{\text{HLA}[i]} + \beta_{\text{class}[i]}\right) \quad (1)$$

Here, $p_i$ is the proportion of positive peptides for measurement i. The parameter $\beta_0$ is the intercept, corresponding to the overall, average proportion of peptides across HLA-alleles and person-classes (regression parameters in this model fulfill sum-to-zero constraints meaning they are deflections from the overall mean). The parameter $\beta_{\text{HLA}[i]}$ is the influence of the specific HLA allele that is present in measurement i, and $\beta_{\text{class}[i]}$ is the influence of the specific person class for measurement i. The number of positive peptides that is observed in the sample, $n_i$, is then assumed to be drawn from a binomial distribution with the given $N_i$ (number of peptides tested) and the estimated $p_i$:

$$n_i = \text{binomial}(N_i, p_i) \quad (2)$$

From each regression coefficient parameter value in the MCMC sample file we can now compute a predicted population proportion for each class that is corrected for the HLA effects of HLA alleles:

$$p_{\text{class}} = \text{logistic}\left(\beta_0 + \beta_{\text{class}}\right) \quad (3)$$

From the predicted, HLA-corrected proportions we can furthermore compute any derived measure of interest, including the log fold change measure that we report in this study:

$$\log \text{fold change} = \log\left(\frac{p_{\text{class1}}}{p_{\text{class2}}}\right) \quad (4)$$

Notice that we automatically account for uncertainty in these measures since they are computed for each value in the sample file for the regression coefficients, resulting in posterior distributions for the derived measures also. We can therefore compute, e.g., the probability that log fold change is larger than zero, corresponding to the probability that $p_{\text{class1}} > p_{\text{class2}}$, i.e., that the HLA-corrected proportion of recognized peptides is larger in class 1 than in class 2.

*Bayesian model normalizing HERV response to viral response (Fig. 3g).* We also used the response to the set of viral peptides as an internal control for the overall level of immune response. Specifically, we, for each class of individuals, obtain a posterior distribution over the proportion of viral peptides recognized in that class of people (see third model mentioned above). From the posterior distribution of the HLA-corrected proportion of HERV peptides recognized ($p_{\text{HERV, class, HLA-corrected}}$), and the posterior distribution of the proportion of viral peptides recognized ($p_{\text{viral, class}}$),

we then compute the posterior distribution of their ratio:

$$p_{\text{class, normalised}} = \frac{p_{\text{HERV, class, HLA-corrected}}}{p_{\text{pviral, class}}} \qquad (5)$$

From the joint posterior distribution over these normalized values we can finally compute the probability of contrasts of interest (the probability that the HLA-corrected, normalized proportion of HERV peptides is larger in some classes than in others).

*Bayesian model of the connection between clinical outcome and T cell response to HERV or viral antigens (Fig. 4d).* We explored the connection between clinical outcome (where patients are divided into responders or non-responders) and the combined effects of the responses to HERV and viral antigens. In this model predictors were: (1) indicator variables for the 4 HLA alleles, (2) indicator variable for whether there was any HERV response, (3) indicator variable for whether there was any viral response, and (4) the interaction between response to HERV and viral peptides (i.e., the effect of having responses to both):

$$p(\text{responder}) = \text{logistic}(\beta_0 + \beta_{\text{HLA}}x_{\text{HLA}} + \beta_{\text{HERV}}x_{\text{HERV}} + \beta_{\text{VIR}}x_{\text{VIR}} + \beta_{\text{HxV}}x_{\text{HERV}}x_{\text{VIR}})$$
$$(6)$$

We again used hierarchical priors to regularize estimates and avoid overfitting: here the three coefficients for HERV, VIR, and HERVxVIR were assumed to be drawn from a common normal distribution.

**Reporting summary**. Further information on research design is available in the Nature Research Reporting Summary linked to this article.

## Data availability

The data that support the findings of this study are available from the corresponding author upon reasonable request. HERV peptides affinity to MHC class I molecules was predicted using NetMHCpan version 2.8 (http://www.cbs.dtu.dk/services/NetMHCpan-2.8/). Cancer testis antigen (CTA) library was developed using data available on Ctdatabase (http://www.cta.lncc.br). Accession numbers for known transcribed HERVs published by Mayer et al. were used to download the reported nucleotide sequences from the National Center for Biotechnology Information's (NCBIs) Entrez database (DB, https://www.ncbi.nlm.nih.gov/Web/Search/entrezfs.html). Source data are provided with this paper.

## Code availability

Codes used for data analysis are available on GitHub (https://github.com/SRHgroup/HERV).

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

## Acknowledgements

We thank all patients and healthy donors for contributing the analyzed samples. We thank all collaborators for their active participation in this work, specifically Bente Rotbøl, Anna Gyllenberg Burkal, Julien Candrian, Anni Flarup Løye, and Anja Holm Vestergaard for excellent technical support. This work was supported by grants from the European Research Council (StG 677268 NextDART to S.R.H.), the Danish Cancer Society (R124-A7695 to K.G.), The Lundbeck Foundation (R190-2014-4178 to S.R.H., R181-2014-3828 to S.K.S., and R180-2014-3675 to A.D.Ø.), the Novo Nordisk Foundation (NNF13OC0003435 to K.G.), and the Van Andel Research Institute through the Van Andel Research Institute—Stand Up to Cancer Epigenetics Dream Team (to S.R.H., K.G., and S.K.S.). Stand Up to Cancer is a division of the Entertainment Industry Foundation, administered by AACR. National Health and Medical Research Council (Australia) (APP1163815 to A.U., and APP1024364, 1043934, and 1102589 to J.P.), Leukemia & Lymphoma Society (USA) (to A.U.), Cancer Institute of New South Wales/Translational Cancer Research Network (to J.P.), and Anthony Rothe Memorial Trust (to J.P. and A.U.).

## Author contributions

S.K.S., A.D.Ø., and A.M.B. conceived the idea, designed research, performed experiments, analyzed data, prepared figures, and wrote the paper. A.U. and J.P. designed research and provided samples from the Australian patients. S.H.-T. performed experiments for T cell analysis. A.B. performed bioinformatics analysis on RNA-seq data and prepared figures. K.V.J. and G.A. performed bioinformatics data analysis. A.K.B. and A.M.M. assisted in barcode experiment and data evaluation. T.T. prepared MHC class I monomers. A.O.G., I.H.D., A.D.Ø., M.B.T., and K.G. collected and processed samples from the Danish patients. A.G.P. performed Bayesian analysis, statistical analysis, and wrote the paper. N.T. conducted mass-spec analyses and data revision. Z.S. and A.C.E. conceptualized and supervised HERV analysis. K.G. conceived the idea and supervised the study. S.R.H. conceived the idea, supervised the study, analyzed data, and wrote the paper. All authors revised and approved the final version of the paper.

## Competing interests

K.G. has served as advisory board member for Celgene. The remaining authors declare that they have no competing interests.
