## [Peer Review File · Nature Communications]

Reviewers' comments:

Reviewer #1 (Human endogenous retrovirus, cancer, systems biology)(Remarks to the Author):

In their manuscript, Sairi SK et al described that HERVs form a reservoir of T cell targets in hematological malignancies with low mutational burden.

The author used an elegant approach using HERV, and CTA (cancer testis antigen) and antigens from common viruses (EBV, CMV, flu, ...) as controls. Overall 18 out of 66 (at last 49) picked HERV were found to produce 29 out of 1169 HERV-derived peptides recognized by CD8+ T cells in half of 34 patients suffering from myeloid malignancies. The authors demonstrated that T cell recognition of HERV-derived elements is present in myeloid malignancies, and claimed a disease-specific enrichment of such CD8+ T cells. They also shown that HERV-specific T cells are significantly enriched in patients, and that the presence of such HERV-specific T cells strongly correlated with the expression level of HERVs. At this point, the impact of the AZA treatment (responders versus non-responders) on the frequency of HERV-reactive T cells and a mediated immune reconstitution, although exciting, appears overestimated as not supported by statistically significant data. The manuscript is sometimes difficult to follow, including in my opinion some unnecessarily complex results and a discussion too far from the presented results.

In addition, although I am not a specialist of statistics, I am disturbed by the way some results are illustrated and analysed (e.g. fig 3, fig 4, fig S2). The author used bar plot without systemically specifying if they represent the median or the mean. In addition they seem to show SD rather than 95% CI. It seems to me that a representation in box plot would be more informative. Concerning the statistical analysis (fig legends 3 and 4) it is indicated that "an unpaired t test(Mann-Whitney test)" was used. The unpaired t test is a parametric test which compares the means of two unmatched groups, assuming that the values follow a Gaussian distribution and that the two populations have the same variance. If this is not the case, the nonparametric Mann-Whitney test (unmatched groups, healthy versus patients) or the Wilcoxon test (paired groups, before / after treatment) should be used.

MAJOR COMMENTS

Introduction

P3: Rather than citing a group of reviews (4-6), it would be more informative to cite some clinically relevant studies showing significant expression of defined HERV loci in different cancers (testis, colon, breast) and keep the recent review on implication for immunotherapeutics of cancer (7)

P4: ... which can promote tumor-specific recognition by T cells in vivo 19–27. While the Balada review is excellent, this is off topic. It would be better to cite HERV/T cell response in different solid tumors (e.g. seminal work of Mullins CS HERV-H in in colon cancer, Cancer Immunol Immunother 2012, in addition to HERV-E in kidney, HERV-K in breast and seminoma, and obviously HERV-K-MEL in melanoma)

Results

P5: "Results section, Prediction of HERV-derived T cell antigens"

HERV DEFINITION : The aim of this work is not to identify which HERVs may induce an optimal CD8+ response in haematological malignancies but to demonstrate that HERV may represent a reservoir of T cell targets in haematological cancers. This is not new in the cancer field, but this was not demonstrated before in myeloid malignancies.

Nevertheless, there is a lack of description of the 66 selected HERV elements, to which group do they belong (is there any bias toward some groups, or regarding the complexity of the HERV part of our genome); do they contain gag pol env sequences, in which tissues (PBMCs, cancer cells, testis, placenta, other) transcription have been previously identified, is the transcription driven by LTR or cellular promoter (to support or not a generalization of the observation) and by the end to which protein(s) correspond the CD8+ reactive peptide?

P7: "Results section, T cell recognition of HERV-derived elements is present in myeloid malignancies"
COHORT DEFINITION: A clarification of the definition of cohorts for each experiment would be helpful:
- ... to further evaluate if HMA therapy enhances the T cell reactivity to our selected group of HERVs.

Two patient cohorts with samples collected ... "two patients cohorts" The wording "two patients cohorts" is used many time with apparently different meaning

\p7 cancer patients from Rigshospitalet plus Herlev Hospital (what is the difference between RH and HH?) and patients from SH (Sydney cohort)? This last cohort was previously published (ref 35) do you use all the cohort? Did you select individuals (if so based on which criteria?)?, see legend figure 5 et RNA-seq data analysis ..) what means SH. By the end, is it relevant to pool all the patients?

\p12 same as in p7 (mix of 3 sites and 2 conditions PBMCs and BM)

\P16 two cohorts appears to define "healthy donors and patients" but it not clear which patients. (RH, HH, SH, pbmc or bone marrow?)

So you have three sites, two types of samples (PBMCs and BMCs), two modalities of AZA treatment and PBMCs from healthy individuals. Please justify the rational of patients/samples aggregation when you compare two conditions.

P12: "Results section, HERV-specific T cells are significantly enriched in patients"

Overall, if the conclusion "In summary, we observed a significant amount of HERV-reactive T cells in patients with hematological malignancies." sounds reasonable, the second statement "AZA treatment further increased HERV-reactive T cells in these patients." appears poorly evidenced.

See comment above with P7. The comparison of PBMCs + BMCs patient samples with PBMCs of healthy donor is not very consistent in my opinion; you should better compare PMBCs (or argue substantially why it make sense to mix populations)

I do not see why the authors state " AZA treatment further enhanced the frequency of HERV-reactive T cells compared to before treatment samples " as no statistical differences are observed between both conditions (fig3A); it is not the case indeed in figS4 (reactive T cells: 8 up ,7 down, 2 unchanged). Does it reflect the absence of effect of AZA or a disparate effect that would require additional criteria to stratify patients? This should be clarified. The number of samples would need to be increased to see if the treatment actually has an effect.

The same comment applied to CTA analysis; there is no statistical results supporting the assertion of an AZA effect. No statistical data are provided with "patient-specific enrichment of T cells reactive to CTAs", what makes weak the statement "patient-specific enrichment of T cells reactive to CTAs" and thus the pooling of HERV and CTAs appears irrelevant as far as I understand 'fig3C). The argument "As both HERV and CTA genes are epigenetically regulated" is very weak as epigenetic mechanisms may be distinct for HERV and CTA and indeed among HERV elements (DNA methylation, histone modifications RNA-and polycomb-based mechanisms, miRNA, ...)

Nevertheless, in my opinion, the main question concerns the enrichment of HERVs specific T cell in patients; not the impact of AZA. I would suggest that it could be more relevant in this paragraph to compare HERV-specific T cells enrichment in patients (fig 3A) with CTAs (fig 3B) and viral antigens (figA and B corresponding to middle and right in fig3) to appreciate the bias (or not) behaviour of T cell response to HERV antigens

P16: "Results section, Presence of HERV-reactive T cells associates with therapy-mediated immune reconstitution"

If I understand correctly the message, the main question is before/after treatments by dichotomizing the patient population in responders/non"responders (and not healthy/patients)

P16-17: On line with the proposed above simplification, I suggest to group the results addressing this aspect, from fig 3D left and middle with fig4D and E, highlighting the differences in HERV, CTAs, Viral antigens in responders non responders, even if those results do not appears statistically significant (due to the size of the sampling? See the third comment related to P12) in any conditions comparing

"before" and "after". At first glance, the overall shape appears similar for HERVs and viral antigens (although at different level) , but different for CTAs

P17: The only significant (and intriguing) observation concerned AZA effect in responders patients with HERV-derived T cell reactivity (fig4E); it should be of interest to see if the same behaviour occurs with CTAs (viral antigen reactive T cells in CTAs positive samples), to appreciate the potential added value (or not) of HERV-derived cell recognition

P18: "Results section, Transcription of HERV elements is associated with malignancy"

This part is very complex and confusing. I suggest that the authors should focus on showing first overexpression of HERVs in patients (fig 5c), then illustrate the correlation between expression and T cell response (5A, 5B?), third, illustrate the influence of AZA treatment on HERV expression AND associated T cell response. Particularly, is there a correlation between the level of expression of a specific HERV locus in a patient and the presentation of the associated peptides in the same patient?

P18. The authors indicate that "across all the HERVs evaluated for T cell recognition we found that transcript levels were higher (median=2.06) for those recognized by T cells compared to those not recognized (median =1.19) (Fig. 5B)" and in the figure legend, the associated p value of 0.34 does not indicate a significant difference. This should be clarified

P19 "may explain the limited AZA-mediated enhancement of T cell recognition following treatment". It was previously indicated (P13) that "AZA treatment further increased HERV-reactive T cells in these patients. ". I must admit I am a little bit confused by this apparent contradiction. Please clarify.

P19 and p20:Fig S6 and 7 and the associated text in the main manuscript does not provide any information or understanding about HERV regulation and appears off-topic.

P21 In fig5D, each individual HERV shared globally a similar regulation among patients (generally up or down regulated), the situation appearing more complex as regard to AZA treatment (up AND down regulated = f(patient)); this could be discussed.

DISCUSSION

The discussion is too far from the results presented, and therefore both general and speculative. The authors should focus on their specific results (is it specific)

-The authors cite HERV ERVFRD-1, ERVFRD-2, ERVE-3, and several members of ERVK family to be primarily T cell 'immunogenic'. Is there any bias regarding the initial selection of HERV elements?

-The authors claimed a disease-specific enrichment. It would be interesting to compare their peptides with those of the literature. First, does their catalogue of peptides show peptides identified as active in other contexts than haematological malignancies (eg colon cancer, ref 20 and 21, 39 see Krone B European Journal of Cancer 2005, other cancers). Second, the 29 peptides identified as reactive in hematological cancers are they also active in other contexts?

-The impact of AZA treatment should be carefully discussed

Minor comments

It would be helpful to include line numbers on the submission to receive more specific instructions and suggestions

P5 "Results section, Prediction of HERV-derived T cell antigens"

-HERV loci ... originate from Repbase. This is untrue. Repbase contains prototype sequences, generally consensus sequences of large families of repeat.

-Explained why 1,169 out of 1,173 peptides were synthesized (in mat meth?)

-the binding rank score should be better explained (in mat meth and in legend of table S1) for those not familiar with it.

P7: "Results section, T cell recognition of HERV-derived elements is present in myeloid malignancies"

P8 Epstein-Barr not Ebstein-Barr (typo error)

P8/sup: What is the meaning of the 2 grey boxes on the log2fc scale (overlap grey and red for ???) in fig S1 (also in figure 2)

P8/sup: I am not convinced by fig S2 which is supposed to illustrate that reactivity is similar in PBMCs and BMCs. No responders(0 = no response) should be excluded; this induce a bias in the statistical tests. A box plot with the outliers would be more informative (see general comment above).

P8: Figure 2 and 2S and other: A and B labelling for after and before AZA treatment is confusing with figure enumeration A,B,C, D,E. Explicitly indicate after and before.

P8: ... 18 out of 49)... The transition between the number of HERV-derived epitopes (29 (out of 1,169) and HERVs loci (18 out of 49) can be hard to follow. Please indicate explicitly that the 29 peptides correspond to the 18 HERV loci.

P12: "Results section, Transcription of HERV elements is associated with malignancy"

P17 Fig 4C appears irrelevant regarding "therapy-mediated immune reconstitution"

Why is the RNA seq information not provided in the mat meth section? (tab S5)

P21: In fig 5c, patients (fig 5c; 49 HERVs x 18 patients =882 points; is it correct? please clarify in the legend). In fig5 legend : Fig 5 E similar to C?? didn't you mean D. The information "responder / no responder" could be of interest to tag patients.

P27 the dose of AZA used in patients from Australia is not indicated

P29 Cancer testis antigen: there is an error in the numbering of S2 and F3 tables, actually S3 and S4

P30 DNA barcode: attached to PE ... in full please

P33 RNA-seq data : words are missing in the two before last sentences (the comparison of healthydata are included)

P28 table S2 sup data. Column IPSS-R or CPSS. What means INT-1, INT-2, is "high" similar to "good" in the cytogenetics-IPSS; please clarify

Reviewer #2 (Leukemia, anti-cancer immunity)(Remarks to the Author):

The manuscript entitled "Human endogenous retroviruses form a reservoir of T cell targets in hematological cancers with low mutational burden" by Saini et al. investigates the potential of polypeptides derived from human endogenous retroviruses (HERVs) in targeted immunotherapy. Although broadly distributed in the human genome, epigenetic mechanisms prevent the expression of HERV-encoded genes as the expression of these genes would mimic viral infection leading to elimination of the respective cells.

In the current manuscript, the authors examined whether HERV-encoded elements activated in myeloid malignancies might serve as targets for immunotherapy. To this end, the authors used a peptide-MHC-multimer-based technique to specifically identify reactive T cells. They used the four most abundant HLA alleles of Caucasian populations together with a peptide-library consisting of 1,169 potential antigenic peptides derived from 66 HERV genes. Using this technique, the authors identified 29 HERV-derived peptides as potential targets for recognition by CD8+ T cell. Moreover, the authors detected T cell recognition of HERV-encoded peptides in 17 of 34 examined patients.

In summary, the study supports available data on HERV transcription and the role of methylation modifying drugs in cancer treatment. However, despite the elegant study with state-of-the-art methods, the claim that anti-HERV-T cells could be exploited in immunotherapy is an overstatement due to a lack of homogenous T cell epitopes throughout the patient cohorts. Additionally, HERV-reactive T cells could not be detected in all patients that underwent AZA treatment or are suffering from myeloid malignancies.

Specific Comments:

1. HERV envelope mRNA presence does not necessarily mean that proteins are translated due to inhibitory factors such as RIG-I and Apobec3. Furthermore, HERV proteins might not form infectious HERVs and might not be recognized and subjected to MHC-I presentation in vivo. It is known that TLR-7 plays a crucial role in HERV depression in vivo. Can the authors provide direct evidence for the expression of some of these peptides in cancer cells?
2. Do the patients have detectable and infectious HERVs in peripheral blood (In vitro infection assay)?
3. Is the overall T cell immunogenicity (anti-viral) as shown in Figure 4 related to enhanced anti-viral factor expression such as Interferon- β , IFN- γ , IL-12, TLR-7?
4. The authors show in Figure 2 and Figure 3 that HERV recognizing T cell numbers are increased after AZA treatment. Do these patients show increased levels of anti-viral factors/cytokines in peripheral blood? The non-responders to AZA-treatment show the largest increase in HERV-reactive T cells after treatment (Figure 3D, right). Is this not in sharp contrast to the hypothesis that HMAs treatment activates HERVs which then lead to T cell activation and subsequent elimination of HERV-positive cells?
5. Can the authors directly show that anti-HERV T cells are activated by HERV:MHC-I presenting myeloid cancer cells?
6. Figure 4, the link between AZA treatment, activation of HERVs and general improvement of immunological responses is unclear. Is there a significant difference in T cell numbers reactive to common viral antigens before and after treatment? Is this the consequence of general T cell activation directly induced by AZA treatment or are common viral antigens expressed by HERVs?
7. Would the inclusion of more HERV peptides provide a more homogenous pattern of anti-HERV T cells?
8. Page 10, Figure 2: Abbr. for before (B) and after (A) is misleading. Healthy donors show anti-HERV T cells similarly to patients within the allele groups of HLA-A01:01, B08:01. Furthermore, there is no homogenous pattern of HERV-reactive T cells to certain peptides as in the viral group of B08:01. A homogenous pattern might be needed for a therapeutic approach.
9. Page 8, line 4: change "Ebstein-Barr" to "Epstein-Barr".

Human endogenous retroviruses form a reservoir of T cell targets in hematological cancers with low mutational burden

Reviewer #1 (Human endogenous retrovirus, cancer, systems biology)

The manuscript is sometimes difficult to follow, including in my opinion some unnecessarily complex results and a discussion too far from the presented results.

In addition, although I am not a specialist of statistics, I am disturbed by the way some results are illustrated and analysed (e.g. fig 3, fig 4, fig S2). The author used bar plot without systematically specifying if they represent the median or the mean. In addition they seem to show SD rather than 95% CI. It seems to me that a representation in box plot would be more informative.

Concerning the statistical analysis (fig legends 3 and 4) it is indicated that “an unpaired t test (Mann-Whitney test)” was used. The unpaired t test is a parametric test, which compares the means of two unmatched groups, assuming that the values follow a Gaussian distribution and that the two populations have the same variance. If this is not the case, the nonparametric Mann-Whitney test (unmatched groups, healthy versus patients) or the Wilcoxon test (paired groups, before / after treatment) should be used.

First, thank you for a very thorough and thoughtful revision of our manuscript. We appreciate your efforts and valuable suggestions that allow us to improve the manuscript. In the revised version of the manuscript, we have incorporated your comments and suggestions and we are hopeful that you will find this version with good scientific quality, clear data representation, and simpler reading.

Thank you for detailed comments on data presentation and statistical analysis. We apologize for not providing sufficient information for each of the statistical analyses performed on our data set in this manuscript. And we agree with you that it would be more informative to use box plots to illustrate the data that we presented as bar plots in the previous version of the manuscript.

As the statistical evaluation and representation is central for our findings, in the revised version we have put substantial efforts to evaluate the data using well accepted statistical procedures and presented them in a meaningful way providing all the detailed statistical output in the main text and/or in the figure legends. Detailed description of each statistical analysis has been included under the “statistical analysis” in the materials and methods section.

Furthermore, Prof. Anders Gorm Pedersen, who is professor of bioinformatics and have worked extensively on statistical data modeling and patient data statistics, has been involved to evaluate and overlook statistical analysis shown in the revised version. This allowed us to analyze and illustrate our data using robust statistical models, write interpretations supported

by appropriate statistical evaluation, and illustrate the findings more clearly. The analyses and modifications for each individual figure/data set are elaborated in the following section:

Statistical analysis and data illustration:

As you rightly explained, if a data set does not follow a Gaussian distribution a nonparametric t test would be more appropriate for statistical comparison of two data sets. We have tested all data sets for a Gaussian distribution using normality test by GraphPad Prism data analysis software. Data sets (revised version) in Fig. 3 (B and D), Fig. 4 (D-F), Fig. 5 (A and C), Fig. S3A (Fig. S2 of old version), Fig. S3C, Fig. S5 (A-C), and Fig. S6 (B-C) do not follow a Gaussian distribution. We missed to state this information in the previous version of the manuscript. Therefore, all above mentioned data sets are analyzed using a nonparametric t test. We used the Mann-Whitney t test to compare unmatched groups and Wilcoxon matched-pairs signed rank test for paired analysis and we have used box plots to illustrate these data sets.

Figure-specific details:

Fig. 3:

- Panel B and E (middle panels in the previous version): Bar plots replaced with box plots. Healthy donors and patient samples were analyzed using nonparametric Mann-Whitney t test, and pre-AZA and post-AZA patient samples were analyzed using Wilcoxon nonparametric paired t test.
- Panel A and D (left panels in the previous version): Bar plots replaced with eye plots illustrating individuals with T cell reactivity in each group as proportions, showing median value for each group with 90% credible interval (CI). Statistical comparisons were made using a statistical model calculating probability (in %) by which a group is different with the compared group. We have stated the median with the CI values (lower and upper) and probability values in the figure legend. The details of the statistical model is explained in the methods section.
- Panel C and F: Fold change in T cell reactivity (data sets shown in panel B and E) in patients compared to healthy donors, and in post-treatment compared to pre-treatment. A regression model is employed taking into account the HLA coverage in each individual sample, and potential HLA-driven differences in T cell detection. Fold change was calculated by dividing the HLA allele corrected proportions of T cell reactivity for individuals in a group of samples with the similar proportion of the compared group using this regression model, and probability score was calculated to show the positive fold change (>0). The plots show the log fold change median value with 90% CI. We have stated the median with the CI values (lower and upper) and probability ($X > 0$, in %) for all compared data sets in the figure legend as well as in the text wherever found necessary. The detailed regression model is explained in the methods section.
- Panel G: Fold change in HERV-reactive T cell reactivity (data sets shown in panel B) using a regression model. This statistical model further considers viral antigen-specific T cell reactivity (based on the reduced viral antigen-specific T cell reactivity shown in panel E) as a covariant over the HLA correction applied in panel C. This analysis was performed to understand the HERV-driven T cell reactivity in the context of reduced T cell reactivity

observed in the patient cohort (Fig. 3D-E). The plots show the log fold change median value with 90% CI. We have stated the median with the CI values (lower and upper) and probability ($X > 0$, in %) for all compared data sets in the figure legend as well as in the text wherever found necessary. The detailed regression model is explained in the methods section.

Fig. 4:

- Panel D-F. Bar plots replaced with box plots. Statistical analysis was performed using Wilcoxon nonparametric paired t test (for D and E) and Mann-Whitney nonparametric unpaired t test (for F). For data sets in panel F an unpaired t test was done to compare pre- and post-AZA patients as only individuals with or without HERV-specific T cell responses were segregated in the analysis, hence data are not paired.

Fig. 5:

- Panel A: Mann-Whitney nonparametric unpaired t test was used to compare expression levels in healthy donors with expression levels in patients (pre- and post-AZA) and Wilcoxon nonparametric paired t test to compare pre- and post-AZA group. The analysis was performed using R data analysis software.
- Panel B: Spearman correlation using GraphPad Prism.
- Panel C: Violin plots replaced with box plots, data sets were compared using Mann-Whitney nonparametric unpaired t test.

MAJOR COMMENTS

Introduction

P3: Rather than citing a group of reviews (4-6), it would be more informative to cite some clinically relevant studies showing significant expression of defined HERV loci in different cancers (testis, colon, breast) and keep the recent review on implication for immunotherapeutics of cancer (7)

Thank you for your suggestion. We have changed the references so that they now include original studies showing disease-specific overexpression of different HERV loci. Regarding autoimmune diseases we have included ref. 4 showing increased RNA level of a specific HERV in multiple sclerosis. Regarding cancers we have included ref. 5-11 showing increased levels of defined HERVs in testicular, ovarian, breast, prostate, melanoma, and colon cancer. Besides these we have also included ref. 12 (Rooney et al.) here, as this study demonstrated the enhanced transcription of many of our chosen 66 HERVs across a number of tumor types. Moreover, we have kept a recently review from our group on the potential immunogenic role of HERV transcription (ref. 13, Attermann et al.).

P4: ... which can promote tumor-specific recognition by T cells in vivo 19–27. While the Balada review is excellent, this is off topic. It would be better to cite HERV/T cell response in different solid tumors (e.g. seminal work of Mullins CS HERV-H in colon cancer, Cancer Immunol Immunother 2012, in addition to HERV-E in kidney, HERV-K in breast and seminoma, and obviously HERV-K-MEL in melanoma)

Thank you for your observation. We have removed the Balada review, which obviously does not fit here. We have also removed the review by Kassiotis and Stoye. Instead we have focused only on original studies including your pertinent suggested study by Mullins et al. (ref. 31).

Results

P5: “Results section, Prediction of HERV-derived T cell antigens”

HERV DEFINITION: The aim of this work is not to identify which HERVs may induce an optimal CD8+ response in haematological malignancies but to demonstrate that HERV may represent a reservoir of T cell targets in haematological cancers. This is not new in the cancer field, but this was not demonstrated before in myeloid malignancies.

Nevertheless, there is a lack of description of the 66 selected HERV elements, to which group do they belong (is there any bias toward some groups, or regarding the complexity of the HERV part of our genome);

There is definitely a bias towards the HERVK group in the HERVs found to be transcribed. We have edited the text accordingly (page 5, line 8-9):

“66 HERVs originate from 17 different groups, annotated according to Repbase, and of these the HERVK group has the highest representation (39%).”

do they contain gag pol env sequences,

We were not able to receive this information for all the HERVs involved in our T cell screening. For some, this information was available for original references, but not for all. Hence we can unfortunately not provide a full overview related to the content of such sequences.

in which tissues (PBMCs, cancer cells, testis, placenta, other) transcription have been previously identified,

We have generously been provided with the data behind the supplementary data figure 4SE of the Rooney et al paper, Cell, 2016 (ref. 12). Here they had analyzed the same 66 HERVs in the

entire TCGA as well as GEx data bases. From these data we have generated a new supplementary figure (Fig. S1) showing the expression of 57 of the 66 HERVs having more than 1 read per million (RPM).

The manuscript text has been edited accordingly (page 5, line 9-14):

“The 66 HERVs are generally transcribed at very low levels in the majority of the body’s healthy tissues as well as in cancers (Fig. S1). The exception to this is ERVV-2, ERVV1, ERVK-6, ERVFRD-2, ERVFH21-1 all highly transcribed across both normal and cancerous tissue. Some HERVs are also tumor and tissue specific, for example ERVH-5, ERVH48-1 and ERVE-4 shown by Rooney et al to be tumor specific.”

is the transcription driven by LTR or cellular promoter (to support or not a generalization of the observation) and by the end to which protein(s) correspond the CD8+ reactive peptide?

We have not been able to receive this level of information for the majority of the HERVs analyzed. We have focused on evidence for translation of HERVs, as this is a demand for potential T cell reactivity. Obviously, understanding their transcriptional regulation could provide further knowledge to whether this parameter may play a role for T cell recognition.

P7: “Results section, T cell recognition of HERV-derived elements is present in myeloid malignancies”

COHORT DEFINITION: A clarification of the definition of cohorts for each experiment would be helpful:

- ... to further evaluate if HMA therapy enhances the T cell reactivity to our selected group of HERVs. Two patient cohorts with samples collected ... “two patients cohorts “ The wording “two patients cohorts” is used many time with apparently different meaning

▫p7 cancer patients from Rigshospitalet plus Herlev Hospital (what is the difference between RH and HH?) and patients from SH (Sydney cohort)? This last cohort was previously published (ref 35) do you use all the cohort? Did you select individuals (if so based on which criteria?)?, see legend figure 5 et RNA-seq data analysis ..) what means SH. By the end, is it relevant to pool all the patients?

▫p12 same as in p7 (mix of 3 sites and 2 conditions PBMCs and BM)

▫P16 two cohorts appears to define “healthy donors and patients” but it not clear which patients. (RH, HH, SH, pbmc or bone marrow?)

So you have three sites, two types of samples (PBMCs and BMCs), two modalities of AZA treatment and PBMCs from healthy individuals. Please justify the rational of patients/samples aggregation when you compare two conditions.

Thank you for your comment. We admit that the definitions were too complicated. We have now simplified this and edited the manuscript according to this:

- The patients are now collectively referred to as the patient cohort (n=34). This cohort consists of two groups: The Danish patients (n=22) and the Australian patients (n=12).
 - This means that the wording “the two patient cohorts” is now omitted.

- The group of healthy donor controls are now simply referred to as the healthy donors (n=27).

Regarding your sub-comments:

1. The designations, RH and HH, are now omitted. RH and HH were shortenings of Rigshospitalet and Herlev Hospital, which are two hospitals in Copenhagen, Denmark. They are both part of Copenhagen University Hospital and the patients from these two hospitals have been treated the same way.
2. The former “SH cohort” is the Australian patients, which were enrolled in New South Wales (incl. Sydney), Australia. We previously published a study based on analyses on samples from these patients (Unnikrishnan et al., *Integrative Genomics Identifies the Molecular Basis of Resistance to Azacitidine Therapy in Myelodysplastic Syndromes*, Cell Reports, 2017). For our current T cell analyses we could include samples from 12 of these patients. This was based on sample availability of live frozen MNCs from pre- and post-treatment timepoints as well as a matching HLA type.

- We have added the following sentence in the Materials and Methods section – Patients and sample collection (page 32, line 20-22):

“As for the Danish patients, the Australian patients were selected based on their sample availability before and following AZA treatment and their HLA type.”

With regards to the RNA expression analysis, we were able to include samples from more Australian patients, since we had RNA-seq data available from 18 patients, including the 12 patients also included in T cell analyses.

- We have added the following sentence in the Materials and Methods section – Patients and sample collection (page 33, line 10-13):

“Live frozen BMMCs from 12 Australian patients could be included in the T cell analysis, whereas samples from the total cohort of Australian patients (n=18) could be included in the RNA-seq analysis.”

- And to Results section we have added the second part of the following statement (page 22, line 3-7):

“We analyzed the expression of the 49 HERVs used for T cell evaluation at RNA transcript level in bone marrow (BM) CD34+ hematopoietic stem and progenitor cells (HSPCs) from the Australian cohort of patients in the SH cohort (n=18)⁴¹. Of these 18 patients, 12 patients were also included in the T cell analyses.”

3. The relevance of pooling all the patients:
 - The patients are pooled in order to get as big a cohort of patients as possible. The number of patients treated with AZA are relatively low, and the number of patients from whom cells are viably frozen before and during treatment are even lower. So this patient cohort is quite unique – although we would have liked to increase the

number. Moreover, the inclusion criteria regarding matching HLA type also reduces the number of available samples.

- The patients from Denmark and Australia are all treated with the same drug: 5-azacytidine. However, the dosing schedule is not exactly the same. This reflects the different national clinical guidelines. The Danish patients received 100 mg/m²/day subcutaneously of AZA for five consecutive days on a 4-week cycle according to the Nordic MDS guidelines (<http://www.nmdsg.org>). This regimen varies slightly from the regimen tested in the original clinical trials that led to the approval of the drug by the FDA and EMA. In these trials, AZA was administered at a dose of 75 mg/m²/day subcutaneously for seven consecutive days on a 4-week cycle. This regimen was used for the Australian patients. Although, the total dose of AZA received in each cycle is almost equivalent for the patients, one could speculate that this difference could have an impact on the therapy-induced biological impact. Nevertheless, in terms of patient outcomes there is no evidence of any difference of the two dosing regimens. Thus, in two Spanish studies the overall survival did not differ between the standard 7-day regimen and less intensive regimens (5 days or 7 days with a 2-day break) (Bernal T et al. *Effectiveness of azacitidine in unselected high-risk myelodysplastic syndromes: results from the Spanish registry. Leukemia*. 2015, and García-Delgado R et al. *Effectiveness and safety of different azacitidine dosage regimens in patients with myelodysplastic syndromes or acute myeloid leukemia. Leuk. Res.* 2014).

4. The relevance of pooling PB and BM samples:

- Disease perspective: The malignant cells of myeloid malignancies, as MDS, CMML, and AML, originate from the cells of the hematopoietic system, specifically the hematopoietic stem cells in the bone marrow. However, the malignant cells are circulating both in the bone marrow and in the blood, as almost all myeloid cells in the bone marrow and the blood are part of the malignant clone. Therefore, both T cells in the bone marrow as well as in the blood from patients with myeloid malignancies are considered to be in proximity to the malignant cells. Furthermore, by observation (Fig. S3A), no differences were observed in the level of T cell responses detected in PB versus BM-derived T cell samples. Hence, to provide a sample size relevant for analyses we treated these two sites equally, and merged for further data analyses.
- Clinical perspective: It is important to investigate whether HERV-specific T cells in myeloid malignancies can equally be detected in PB and BM, as it is much easier and patient-friendly to do a PB assessment compared to a BM assessment.

P12: "Results section, HERV-specific T cells are significantly enriched in patients"

Overall, if the conclusion *“In summary, we observed a significant amount of HERV-reactive T cells in patients with hematological malignancies.”* sounds reasonable, the second statement *“AZA treatment further increased HERV-reactive T cells in these patients.”* appears poorly evidenced.

We apologize for this confusion. We have now changed it accordingly (page 14, line 7-9):

“In summary, we observed a significant enrichment of HERV-reactive T cells in patients with myeloid malignancies despite a general immunosuppressive state as observed by decrease in viral reactive T cells in the patients.”

Moreover, every sentence concerning this have been changed in the manuscript throughout.

See comment above with P7. The comparison of PBMCs + BMCs patient samples with PBMCs of healthy donor is not very consistent in my opinion; you should better compare PMBCs (or argue substantially why it make sense to mix populations)

Thank you for the comment. We chose to compare peripheral blood (PB) + bone marrow (BM) from patients with PB from healthy donors (HDs) as we would argue that T cells from BM and PB in patients with myeloid malignancies are part of the same “pool” and circulate between these compartments. This was experimentally supported by our finding of no tissue-specific difference in the detection levels of HERV-specific T cells from PB and BM in the patients (Fig. S3A). Please also see our answer to your question regarding the different cohorts above.

I do not see why the authors state “AZA treatment further enhanced the frequency of HERV-reactive T cells compared to before treatment samples “ as no statistical differences are observed between both conditions (fig3A); it is not the case indeed in figS4 (reactive T cells: 8 up ,7 down, 2 unchanged). Does it reflect the absence of effect of AZA or a disparate effect that would require additional criteria to stratify patients? This should be clarified. The number of samples would need to be increased to see if the treatment actually has an effect.

Thank for your comment. We admit that the statement is confusing. We have now re-analyzed these data sets comparing change in HERV-reactive T cells before and after AZA treatment using a stringent statistical data regression model that corrects for any HLA distribution bias (Fig. 3C) and calculated the probability by which the T cell reactivity increased in post-treatment samples compared to pre-treatment and healthy donors. Therefore, based on this analysis we have edited the specific sentence in the Fig. 3 text accordingly.

Furthermore, when these data were further normalized (considering viral antigen-specific T cell reactivity as a covariant in our regression analysis model) to decreased viral antigen-specific T cell reactivity (Fig. 3G), we did not observe any treatment driven change in HERV-specific T cell reactivity. Therefore, we have added the following text about this observation:

Altogether, it is obvious that we in a follow-up study need to include specific HERVs that are predominantly regulated by methylation and therefore have a high probability of being overexpressed after subjection to AZA therapy in a follow-up study.

The same comment applied to CTA analysis; there is no statistical results supporting the assertion of an AZA effect. No statistical data are provided with “patient-specific enrichment of T cells reactive to CTAs”, what makes weak the statement “patient-specific enrichment of T cells reactive to CTAs” and thus the pooling of HERV and CTAs appears irrelevant as far as I understand (fig3C). The argument “As both HERV and CTA genes are epigenetically regulated” is very weak as epigenetic mechanisms may be distinct for HERV and CTA and indeed among HERV elements (DNA methylation, histone modifications RNA-and polycomb-based mechanisms, miRNA, ...)

Thank you for your comment. We have moved the CTA part to the supplements, because these were not the focus of our study. Therefore, we have also omitted the aggregation of the HERVs and CTAs, as you pertinently suggest. After re-analyzing the data, we do not find a statistical difference between patients and healthy donors regarding CTAs. Therefore, the text has been changed accordingly (page 14, line 3-6):

“Similar to HERVs, we also observed T cells reactive to CTAs in both patients, pre- and post-AZA treatment, and healthy donors (not significant, Fig. S3C). However, these tended to be less frequent and a comprehensive, comparative evaluation would require analysis of a larger cohort.”

Nevertheless, in my opinion, the main question concerns the enrichment of HERVs specific T cell in patients; not the impact of AZA. I would suggest that it could be more relevant in this paragraph to compare HERV-specific T cells enrichment in patients (fig 3A) with CTAs (fig 3B) and viral antigens (figA and B corresponding to middle and right in fig3) to appreciate the bias (or not) behaviour of T cell response to HERV antigens

Thank you for your suggestion. Figure 3 has now been substantially remodeled to focus on the main findings. We have focused on comparing HERV-reactive T cell enrichment in patients compared to healthy donors and used statistical analyses to show that this enrichment is significant and comparing this with reduced viral antigen-reactive T cells in the patients compared to the healthy donors. Hence, we moved the Fig. 4A and B data sets of the old version and reanalyzed the data using the newly developed statistical models together with the HERV-reactive T cell data sets. We have added panel G showing data sets analyzed based on the regression model taking reduced viral antigen reactivity into consideration.

P16: “Results section, Presence of HERV-reactive T cells associates with therapy-mediated immune reconstitution”

If I understand correctly the message, the main question is before/after treatments by dichotomizing the patient population in responders/nonresponders (and not healthy/patients)

Yes, this section concerns the observations in the patients only pre- and post-treatment and split in two groups based on whether or not the patients experienced a clinical response to the treatment. This section has been substantially revised and simplified (Fig. 4D-F), as detailed in the next comment.

P16-17: On line with the proposed above simplification, I suggest to group the results addressing this aspect, from fig 3D left and middle with fig4D and E, highlighting the differences in HERV, CTAs, Viral antigens in responders non responders, even if those results do not appears statistically significant (due to the size of the sampling? See the third comment related to P12) in any conditions comparing “before” and “after”. At first glance, the overall shape appears similar for HERVs and viral antigens (although at different level) , but different for CTAs

Similar to your suggestion for Fig. 3, we have updated the Fig. 4 with the following changes:

We compare pre- and post-AZA HERV- and viral antigen-reactive T cells in responders and non-responders in Fig. 4D and 4E, respectively. HLA-normalized data showed significant enrichment of viral antigen-reactive but not HERV-reactive T cells in responders compared to non-responders in post-AZA samples. Since this was in line with data in Fig. 3E showing immune reconstitution post-AZA treatment, we further show that this effect was observed primarily in patients with HERV-reactive T cells (Fig. 4F).

It would have been interesting to do similar analysis for CTA-positive samples, but our CTA-reactive T cell responses were limited to very few patients, not sufficient to provide any meaningful observations.

In addition to this, we have also included T cell functional data in Fig. 4A-C.

P17: The only significant (and intriguing) observation concerned AZA effect in responders patients with HERV-derived T cell reactivity (fig4E); it should be of interest to see if the same behaviour occurs with CTAs (viral antigen reactive T cells in CTAs positive samples), to appreciate the potential added value (or not) of HERV-derived cell recognition

As mentioned above the limited number of CTA-reactive samples would not allow for such analysis. A larger cohort with specific focus on CTAs would be more appropriate in understanding the impact of AZA treatment on CTAs.

P18: “Results section, Transcription of HERV elements is associated with malignancy”

This part is very complex and confusing. I suggest that the authors should focus on showing first overexpression of HERVs in patients (fig 5c), then illustrate the correlation between expression and T cell response (5A, 5B?), third, illustrate the influence of AZA treatment on HERV expression AND associated T cell response.

We apologize for not presenting the data clearly in this section. We thank you for your suggested changes for a better illustration of the data in Fig. 5. We have now modified Fig.5 accordingly. Panel 5C, 5A, and 5B are now moved to 5A, 5B, and 5C, respectively. Also values of 0 was faulty not plotted in previous version, this has now been corrected in fig 5A.

Particularly, is there a correlation between the level of expression of a specific HERV locus in a patient and the presentation of the associated peptides in the same patient?

Such an analysis would be helpful in showing a direct relation in HERV expression and T cell recognition. We tried to look for a correlation on patient-specific HERV expression and associated T cell recognition. HERV expression data was available for only 12 of the analyzed 34 patients, and of these 12, limited T cell reactivity was observed, and hence not sufficient to evaluate for a potential direct correlation in individual samples. However, a general analysis, across patients, (Fig. 5B) revealed that HERV expression overall correlate well with the T cell reactivity in these patients (Fig. 5B).

P18. The authors indicate that “across all the HERVs evaluated for T cell recognition we found that transcript levels were higher (median=2.06) for those recognized by T cells compared to those not recognized (median =1.19) (Fig. 5B)” and in the figure legend, the associated p value of 0.34 does not indicate a significant difference. This should be clarified

We are sorry for this misleading text. The text has now been corrected (page 22 line 23-24, page 23 line 1-2):

“Furthermore, at a gross level across all the HERVs evaluated for T cell recognition, we found a tendency towards transcript levels being higher for those HERVs recognized by T cells compared to those not recognized.”

P19 “may explain the limited AZA-mediated enhancement of T cell recognition following treatment”. It was previously indicated (P13) that “AZA treatment further increased HERV-reactive T cells in these patients. “. I must admit I am a little bit confused by this apparent contradiction. Please clarify.

We apologize for this. This statement is now drafted appropriately in the revised version and this together with our above-mentioned changes to the statements regarding the AZA-effect on HERV-specific T cells should clarify this former contradiction. The edited text is (page 23, line 17-19):

“This, however, is in line with the observations made in terms of the T cell reactivity to the HERVs, where both pre- and post-treatment samples displayed similar levels of T cell reactivity to HERVs.”

P19 and p20: Fig S6 and 7 and the associated text in the main manuscript does not provide any information or understanding about HERV regulation and appears off-topic.

We agree with your observation. We have excluded the Fig. S7 (related to APM gene analysis) in the revised manuscript. However, since we have identified CTA-reactive T cells in some of the samples, we wanted to look if the expression of these CTAs showed any upregulation similar to what we find for HERVs. Thus, we decided to include the expression data as a supplementary Fig. S7 (S6 of the old version).

P21 In fig5D, each individual HERV shared globally a similar regulation among patients (generally up or down regulated), the situation appearing more complex as regard to AZA treatment (up AND down regulated = f(patient)); this could be discussed.

Thank you for bringing this up. We think it is a very interesting point.

We have added the following section to the discussion (page 30, line 1-15):

“In this study, we found that the investigated HERV loci shared a rather similar regulation at the RNA level across patients before undergoing epigenetic therapy, whereas the expression levels after treatment were more heterogeneous. This may reflect that AZA is incorporated in the DNA of the patients’ malignant cells to different degrees, possibly depending on their division pace which would depend on how dividing they are and on the specific activity of the intracellular enzymes responsible for AZA degradation. Moreover, the activity and efficiency of the intracellular mechanisms responsible for the degradation of HERV RNA molecules could also vary among the patients. Obviously, the relative influence of DNA methylation versus histone modifications on the expression of the specific HERVs may also vary among the patients due to a disrupted cancer epigenome and different evolutionary ages of the specific HERVs^{19,24}. Indeed, as discussed earlier, these specific HERVs may only to a low degree be regulated by DNA methylation. Lastly, the subcellular and clonal composition of the malignant cells will change unequally in the patients after subjection to AZA treatment, demanding single cell analyses.”

DISCUSSION

The discussion is too far from the results presented, and therefore both general and speculative. The authors should focus on their specific results (is it specific)

- *The authors cite HERV ERVFRD-1, ERVFRD-2, ERVE-3, and several members of ERVK family to be primarily T cell 'immunogenic'. Is there any bias regarding the initial selection of HERV elements?*

The HERV elements were not selected based on immunogenicity, but purely based on evidences for transcriptional activity as reported by Mayer et al.

In the revised version of the manuscript we have introduced a T cell reactivity score (Fig. 2F). Based on these findings we have included the following sentence in the discussion (page 27, line 9-12):

"We identified HERVH-5, HERVW-1, and HERVE-3 among the evaluated HERVs to be primarily recognized by T cells, and members of the HERVK family showed a high T cell reactivity score, which may be a consequence of the high expression levels of these HERVs."

- *The authors claimed a disease-specific enrichment. It would be interesting to compare their peptides with those of the literature. First, does their catalogue of peptides show peptides identified as active in other contexts than haematological malignancies (eg colon cancer, ref 20 and 21, 39 see Krone B European Journal of Cancer 2005, other cancers). Second, the 29 peptides identified as reactive in hematological cancers are they also active in other contexts?*

Thank you for bringing this up. We have searched the literature to address this, and found that one of our predicted peptides was earlier found to be T-cell immunogenic in a study including patients with a history of seminoma. However, none of our detected novel 29 peptides have been published as T cell immunogenic in other contexts. The following papers included T-cell reactive HERV-derived peptides: Cherkasova et al., Cancer Res, 2016; Takahashi et al., JCI, 2008; Mullins et al., Cancer Immunol Immunother, 2012; Schiavetti et al. Cancer Res, 2002; Rakoff-Nahoum et al., AIDS Res Hum Retroviruses, 2006; Smith et al., JCI, 2018.

We have therefore added the following section to our discussion (page 27 line 22-23, page 28 line 1-4):

*"Interestingly, one of the HERV-derived peptides included in our library was earlier found to be active in another tumor context (FLQFKTWWI; HLA-A*02:01- and HLA-B*08:01-restricted; HERV-K-derived); thus, CD8⁺ T cell reactivity was detected in a study including patients with seminoma²⁹. However, none of the 29 HERV-derived peptides identified to be active in this study has, to our knowledge, been found active in other contexts."*

- *The impact of AZA treatment should be carefully discussed*

A new section in the discussion addresses AZA treatment and the potential influence on HERV expression and clonal selection (please see the answer to the last comment re results).

Minor comments

It would be helpful to include line numbers on the submission to receive more specific instructions and suggestions

Thank you for your suggestions. Line numbers are now included in the manuscript.

P5 *“Results section, Prediction of HERV-derived T cell antigens”*

- *HERV loci ... originate from Repbase. This is untrue. Repbase contains prototype sequences, generally consensus sequences of large families of repeat.*

Thank you for this observation, which we agree on. This statement has now been deleted.

- *Explained why 1,169 out of 1,173 peptides were synthesized (in mat meth?)*

Custom synthesis of the peptides was feasible for only 1169 of the predicted 1173 peptides due to technical reasons (reported by the manufacturer). We have added this information in the Materials and methods section under “Peptides” (page 35, line 10-12):

“For the HERV peptide library, custom synthesis was successfully achieved for 1,169 out of 1,173 peptides by the manufacturer.”

- *the binding rank score should be better explained (in mat meth and in legend of table S1) for those not familiar with it.*

We have now added a section in the Materials and methods section under “HERV peptide library prediction and selection” describing the percentile rank score and how it is recommended to be used by the authors of NetMHCpan 2.8 (page 34, line 8-16):

“Peptides with a percentile rank score ≤ 2 were selected and included in the final peptide library. This was based on the authors’ guidelines of using netMHCpan 2.8 where peptides determined as binders have a percentile rank score ≤ 2 (weak binders) and ≤ 0.5 for strong binders. The percentile rank for a peptide is

generated by comparing its score against the score-distribution of a large set of random natural peptides. That is a Rank score of 1% means that the predicted score of the peptide is equal to the top 1 percentile score of the large set of random natural peptides. The percentile rank score is recommended, compared to the predicted binding affinity, to ensure predictive compatibility across different HLA alleles.”

*P7: “Results section, T cell recognition of HERV-derived elements is present in myeloid malignancies”
P8 Epstein-Barr not Ebstein-Barr (typo error)*

This is now corrected.

P8/sup: What is the meaning of the 2 grey boxes on the log2fc scale (overlap grey and red for 2??) in fig S1 (also in figure 2)

Figure legend has been updated (page 10):

“T cell responses are shown as log-fold changes based on the baseline input of the complete pMHC library using EdgeR and colored in red scale if significantly enhanced (FDR 0.1%, $p < 0.001$) and grey scale of no significant enhancement was found.”

P8/sup: I am not convinced by fig S2 which is supposed to illustrate that reactivity is similar in PBMCs and BMCs. No responders(0 = no response) should be excluded; this induce a bias in the statistical tests. A box plot with the outliers would be more informative (see general comment above).

We have replaced bar plots with box plots in this figure (new Fig. S3A), and the data sets are analyzed using Mann-Whitney nonparametric unpaired t test (as detailed in the statistical analysis section). We think it would be relevant to show zero values as throughout the manuscript we address the frequency of T cell response detection, and hence also the fraction with no responses should count in. However, we also tried to analyze the data excluding zero values, which showed the same result (shown below).

P8: Figure 2 and 2S and other: A and B labelling for after and before AZA treatment is confusing with figure enumeration A,B,C, D,E. Explicitly indicate after and before.

Thank you for your suggestion. We have now removed the “A” and “B”, and instead used the labels: “Pre-AZA” treatment and “post-AZA treatment”.

P8: ... 18 out of 49)... The transition between the number of HERV-derived epitopes (29 (out of 1,169) and HERVs loci (18 out of 49) can be hard to follow. Please indicate explicitly that the 29 peptides correspond to the 18 HERV loci.

Thank you for your suggestion. The following sentences have now been added/edited accordingly in the abstract and the results section (page 8, line 14-20):

“...we identified CD8⁺ T cell populations recognizing 29 HERV-derived peptides, represented by 18 different HERV loci”

“These 29 peptides derived from 18 different HERV loci.”

“A large fraction of the HERV loci included in this study were found to be immunogenic in patients (18 out of 19; 37%); each giving rise to at least one recognized T cell epitope.”

P12: “Results section, Transcription of HERV elements is associated with malignancy”

P17 Fig 4C appears irrelevant regarding “therapy-mediated immune reconstitution”

The Fig. 4C has now been excluded from the manuscript.

Why is the RNA seq information not provided in the mat meth section? (tab S5)

The RNA sequencing of the samples from the Australian patients was done previously and the applied experimental method was described in our former publication (Unnikrishnan et al., *Integrative Genomics Identifies the Molecular Basis of Resistance to Azacitidine Therapy in Myelodysplastic Syndromes*, Cell Reports, 2017).

Therefore, we have revised the following sentence in the Materials and methods section so that it refers to this publication (page 38, line 18-19):

“Samples from 18 Australian patients were used to generate paired-end RNA-seq data (see ref. 42 for RNA-seq procedure).”

P21: In fig 5c, patients (fig 5c; 49 HERVs x 18 patients =882 points; is it correct? please clarify in the legend). In fig5 legend : Fig 5 E similar to C?? didn't you mean D. The information “responder / no responder” could be of interest to tag patients.

Figure 5C has now become Fig. 5A due to previous request for clarification. And yes, in 5A it is the transcripts from 49 HERVs x 14/18/16 individuals depending on the group. The legend text has been updated (page 26):

“Expression of the 49 HERVs in each of the patients (before; n=18, and after AZA treatment; n=16) compared with healthy donors (n=14); $p < 2.22e^{-16}$ (healthy donors vs patients); $p = 1.8e^{-4}$ (patients before vs after treatment).”

These reference mistake in the legend have also been updated.

Annotations of HERV specific T cell occurrence together with clinical response is annotated in the top bars for each patient in both Fig. 5D and E.

P27 the dose of AZA used in patients from Australia is not indicated

The dose is now indicated in the Materials and methods section (p 32, line 23):

“These patients were treated with AZA at a dose of 75 mg/m²/day s.c. for seven consecutive days on a 4-week cycle...”

P29 Cancer testis antigen: there is an error in the numbering of S2 and F3 tables, actually S3 and S4

We apologies for this error. The references to the table numbers have now been corrected in the revised manuscript.

P30 DNA barcode: attached to PE ... in full please

PE has now been spelled out (p 36, line 9):

“These unique barcodes were attached to phycoerythrin (PE)...”

P33 RNA-seq data : words are missing in the two before last sentences (the comparison of healthydata are included)

We apologize for the missing words, which have now been added (p 39, line 9-13):

“For comparison of healthy donors and samples from patients before treatment, the mean values across all healthy donors were used. For comparison of before and after treatment in patients, the calculated log fold changes were derived from paired samples and only patients with data from both before and after treatment were included.”

P28 table S2 sup data. Column IPSS-R or CPSS. What means INT-1, INT-2, is “high” similar to “good” in the cytogenetics-IPSS; please clarify

The prognostic tools have now been clarified in the legend belonging to Table S3.

Reviewer #2 (Leukemia, anti-cancer immunity)

In summary, the study supports available data on HERV transcription and the role of methylation modifying drugs in cancer treatment. However, despite the elegant study with state-of-the-art methods, the claim that anti-HERV-T cells could be exploited in immunotherapy is an overstatement due to a lack of homogenous T cell epitopes throughout the patient cohorts. Additionally, HERV-reactive T cells could not be detected in all patients that underwent AZA treatment or are suffering from myeloid malignancies.

Thank you very much for your review and for your comments and suggestions.

We have specifically addressed your comment below.

Related to the general concern that T cell responses to HERVs can only be detected in a fraction of patients – it is important to note, that the fraction of patients with such HERV reactive T cells is significantly enriched compared to healthy donors (Fig 3A), even despite the fact that these

patients are often immunocompromised, and hence in general have a lower T cell reactivity (Fig 3D, F, G).

Furthermore, in other studies describing T cell antigens in cancer, a similar picture emerges. Only a fraction of patients have detectable T cell responses to the antigens selected for evaluation^{1,2,3}. This probably relates to internal competition in antigen presentation, HLA diversity, immunodominance etc. Also, our data clearly illustrate that the effect of AZA treatment on HERV expression is highly diverse, and this, we have addressed in the revised discussion. Hence, for immunotherapeutic strategies, HERVs should probably be selected based on personalized gene-expression signatures.

Specific Comments:

1. *HERV envelope mRNA presence does not necessarily mean that proteins are translated due to inhibitory factors such as RIG-I and Apobec3. Furthermore, HERV proteins might not form infectious HERVs and might not be recognized and subjected to MHC-I presentation in vivo. It is known that TLR-7 plays a crucial role in HERV depression in vivo. Can the authors provide direct evidence for the expression of some of these peptides in cancer cells?*

Thank you for your comment, which is a very relevant concern. We have therefore based on a collaboration with Dr. Nicola Ternette from the Jenner Institute, Oxford University, analyzed available mass spectrometry data on “ligandomes” from samples from AML patients and AML cell lines, because such data would strengthen the evidence of actual HERV translation and subjection to MHC I presentation. Indeed, we were able to identify the expression of two of our predicted HERV-derived peptides. An answer to this has therefore been included in the revised version of the manuscript (page 18, line 11):

“In order to understand whether HERV sequences are in fact translated into protein and whether peptides could be presented by leukemia cells in the context of HLA class I, we re-analyzed a published dataset (PMID: 31291378) of HLA-I associated ligandomes of primary AML tumor material and the OCI-AML cell line. We identified peptide sequence TTQEAELLLER, which originates from the ERK3-1 transcript, in two patients (IDs 005686 and 005685). Peptide TEQGPTGVTM, also originating from ERK3-1, was identified in OCI-AML cells. Both of these peptides were part of our HERV library selected for T cell analysis (Table S2). These data further demonstrate that HERV transcripts are expressed, and enter the HLA-I presentation pathway in AML.”

¹ Viborg N, Ramskov S, Andersen RS, et al. T cell recognition of novel shared breast cancer antigens is frequently observed in peripheral blood of breast cancer patients. *Oncoimmunology*. 2019;8(12):e1663107. Published 2019 Sep 30. doi:10.1080/2162402X.2019.1663107

² Andersen RS, Thru CA, Junker N, et al. Dissection of T-cell antigen specificity in human melanoma. *Cancer Res*. 2012;72(7):1642-1650. doi:10.1158/0008-5472.CAN-11-2614

³ Kvistborg P, Shu CJ, Heemskerk B, et al. TIL therapy broadens the tumor-reactive CD8(+) T cell compartment in melanoma patients. *Oncoimmunology*. 2012;1(4):409-418. doi:10.4161/onci.18851

Furthermore, a number of earlier studies have similarly demonstrated T cells targeting HERV-encoded antigens in a variety of cancers:

Cherkasova et al., Cancer Res, 2016: 3 peptides belonging to HERV-E were found immunogenic in clear cell renal cell carcinoma (HLA-A2.01-restricted):

- CR1: FLHKTSVREV; SU1: SLNITSCYV; TM1: LLLQIMRSFV.

Takahashi et al., JCI, 2008: 1 peptide (HERV-E; HLA-A11-restricted) was found in renal cell carcinoma:

- ATFLGSLTWK (named CT-RCC-1)

Mullins et al., Cancer Immunol Immunother, 2012: 3 peptides (from HERV-H Xp22.3; HLA-A2.1-restricted) were found in colorectal carcinoma:

- CLYPFSAFL; PLLSVSLPLL; SLNFNSFHFL

Schiavetti et al. Cancer Res, 2002: 1 peptide (HERV-K; termed HERV-K-MEL; HLA-A*02-restricted) was found in melanoma:

- MLAVISCAV

Rakoff-Nahoum et al., AIDS Res Hum Retroviruses, 2006: 1 peptide (HERV-K; HLA-A0201-restricted) was found in patients with a past history of seminoma:

- FLQFKTWWI

Smith et al., JCI, 2018: 2 peptides (HERV 4700; HLA-A02:01-restricted) were found in clear cell renal cell carcinoma:

- NSWQEMVPV; MVGPWPRPV

2. *Do the patients have detectable and infectious HERVs in peripheral blood (In vitro infection assay)?*

Thank you for your comment. The genetic elements coding for HERVs are impaired and reduced during evolution and as such these HERVs do not form infectious particles anymore. But during transcription they may form dsRNA, that can be recognized by the cellular machinery as viral components, and hence initiate a viral response signature intracellularly.

This has been clarified in the revised introduction of the manuscript (page 3, line 4-8):

“The majority of HERVs are defective due to evolutionarily acquired disruption or silencing mutations and therefore cannot retrotranspose, but instead serve as neutral components of the human genome^{2,3}. Hence, no infectious activity remains from such HERVs, but they may still be recognized as viral components by our immune system.”

3. *Is the overall T cell immunogenicity (anti-viral) as shown in Figure 4 related to enhanced anti-viral factor expression such as Interferon- β , IFN- γ , IL-12, TLR-7?*

Thank you for this interesting comment. Unfortunately, in the current patient cohort, we only have paired RNA seq data and T cell analysis data from 12 of the included patients; hence, prohibiting such further analyses.

4. *The authors show in Figure 2 and Figure 3 that HERV recognizing T cell numbers are increased after AZA treatment. Do these patients show increased levels of anti-viral factors/cytokines in peripheral blood? The non-responders to AZA-treatment show the largest increase in HERV-reactive T cells after treatment (Figure 3D, right). Is this not in sharp contrast to the hypothesis that HMAs treatment activates HERVs which then lead to T cell activation and subsequent elimination of HERV-positive cells?*

Thank you for your comment. We did not find a statistically significant increase in HERV-responsive T cells after AZA therapy in the patients. The data presented in Fig. 3 have been reanalyzed with a regression model taken into account HLA variants across the cohort. We observe a clear difference between healthy donors and patients, but not any further significant increase in the patient cohort after AZA treatment.

Regarding anti-viral factors/cytokines, we would not be able to test this as we do not have blood plasma available from these patients. Furthermore, previous findings suggest that de-repression of HERVs mostly results in intracellular activation of the viral response pathway, rather than true inflammatory signals (ref. 16, 17 and 19 in the manuscript).

5. *Can the authors directly show that anti-HERV T cells are activated by HERV:MHC-I presenting myeloid cancer cells?*

Thank you for this comment. We have tried to address this concern by performing new functional studies including a leukemia cell line and patient material from our patient cohort. Data demonstrating direct recognition of the cell line by HERV-specific T cells are included in the revised manuscript, new Fig. 4A-C. Moreover, the MHC-I peptide elution mass-spec data also concern this issue (see our reply to comment 1).

6. *Figure 4, the link between AZA treatment, activation of HERVs and general improvement of immunological responses is unclear. Is there a significant difference in T cell numbers reactive to common viral antigens before and after treatment? Is this the consequence of general T cell activation directly induced by AZA treatment or are common viral antigens expressed by HERVs?*

The 'viral antigens' that we have evaluated are not HERV derived. All antigens classified as 'Viral' are derived from common viral infections, such as EBV, CMV and FLU. These are fully infectious viruses and raise significantly higher T cell responses than what is normally detected for tumor antigens. The HERV-derived antigens are always determined as 'HERV'. We apologies for a potential confusion between viral antigens and HERV antigens.

Related to figure 4, now D-F in the current version, we observe an increased reactivity to viral antigens in clinical responders versus non-responders. Such increase is also observed when comparing patients pre- to post AZA treatment (Fig. 3E). Thus, we argue that the T cell reactivity to these common viral antigens can be used as a surrogate signal of immunological/hematological improvement. Interestingly, we observed that the increase in viral T cell responses is specifically prominent in the HERV T cell positive fraction of the patient group responding to therapy (Fig. 4F).

7. *Would the inclusion of more HERV peptides provide a more homogenous pattern of anti-HERV T cells?*

It is likely that the inclusion of peptides from additional HERVs would provide a more complete picture of the level of T cell responses to HERVs. Since there is a large variance in the HERV gene elements, and a large fraction of them is likely not translated into protein, we chose to focus on those where most evidence exists for the capacity to be translated.

Furthermore, the T cell immunogenicity of these HERVs seems to differ substantially, which is very clear from our 'T cell reactivity score' (fig 2F). Thus, based on this knowledge, future studies may focus solely on the most T cell immunogenic fraction of the HERVs.

8. *Page 10, Figure 2: Abbr. for before (B) and after (A) is misleading. Healthy donors show anti-HERV T cells similarly to patients within the allele groups of HLA-A01:01, B08:01. Furthermore, there is no homogenous pattern of HERV-reactive T cells to certain peptides as in the viral group of B08:01. A homogenous pattern might be needed for a therapeutic approach.*

Thank you for your comments.

- We agree on the observation and we have now removed the "A" and "B", and instead used the labels: "Pre-AZA" treatment and "post-AZA treatment".
- In the revised version we have included all data split on HLA type (Fig. S4) and we have included a regression model to take into account the HLA variation in the cohort. Data based on this model are presented in the revised version figure 3 C, F and G.

9. *Page 8, line 4: change "Ebstein-Barr" to "Epstein-Barr".*

Thank you for the observation. We have now changed it accordingly.

REVIEWER COMMENTS

Reviewer #1 (Remarks to the Author):

In their manuscript, Saini SK (sorry for the typo in your name in my initial review) et al described that HERVs form a reservoir of T cell targets in hematological malignancies with low mutational burden. They show that HERV-specific T cells are significantly enriched in patients with myeloid malignancies, and that the presence of such HERV-specific T cells correlated with the expression level of HERVs.

In this revised version of the manuscript, the authors have replied to most of my comments and taken into account my suggestions. This is a huge and serious work now clearly presented and simpler to read.

Choice of HERV elements was clarified and explained/illustrated.

As I mentioned in my initial revision, I am not a specialist of statistics. Nevertheless it is clear that the authors used in this revised version robust statistical models and suggest interpretations now supported by appropriate statistical evaluation (AG Pedersen's contribution).

Figures 3 and 4 were largely reorganized and results presented adequately, and figures 1 and 5 were complemented.

References in the text were clarified leading to a better focus of the messages.

The discussion section fits better with the presented results.

Minor comments

page3 lines 5 to 10:

I would not be so affirmative on the fact that HERV cannot retrotranspose (there are HERV RT activities and LINE RT as well) and the wording "serve" as neutral components is unappropriated. In fact (i) "recently" endogenized retroviral elements are often pathogenic, and HERV that persist after negative selection were either (ii) domesticated to serve beneficial roles (eg Syncytins, some LTR enhancers).or (iii) became neutral components of the host genome. Although there are numerous description of viral particles, the key message is that they are no infectious one (a high number of mutations are required to obtain an infectious strain from "nearly" competent sequences, see Phoenix story, Dewannieux M 2006, and MSRV story, Laufer G 2009).

page 6 fig 1

There could be some misunderstanding between the 66 initially selected HERVs and the 49 biologically tested HERVs. This would avoid any possible ambiguity if in figure 1 it was explicitly indicated " final library of 1169 peptides from 49 out of 66 HERVs".

In their reply to my comments, the authors justified why they choose to pool all the patients and also PB and BM samples (reply to my "page 7 major comment", authors points 3 and 4). Obviously, in both situations, it is for sample size reasons/statistical relevance. This nevertheless reveal some limitations of this informative clinical study. A few lines summarizing the limitations of the study and the relevance (and rational) of the choices would be welcome in the discussion section.

Reviewer #2 (Remarks to the Author):

The authors addressed all points appropriately. Obviously, they could not address the question of cytokines etc. to detect inflammation/T-cell activity in this way, as they do not have/could not obtain serum from the donors.

But this does not change the general statement of the paper.

Reviewer #3 (Remarks to the Author):

Comments for authors

I have reviewed statistical aspects of the manuscript and have the following comments:

1. As pointed out by Reviewer 1 in the initial review the T test is a parametric test. There is no such thing as a nonparametric T test, a Wilcoxon t test or a Mann-Whitney t-test. The fact that this was not understood from the comments of Reviewer 1 is a concern. This error must be fixed everywhere it occurs (it is in the figure legends, the Methods section and perhaps elsewhere). It is good that the authors have added a bioinformatician as an author on the revised manuscript to take responsibility for the statistical analysis. It is essential, however, that this author should now review and stand over all aspects of the statistical analysis. My impression is that their contribution to the manuscript has been restricted to the Bayesian analysis.

2. The methods for used for data analysis should be described in sufficient detail in the Methods section and should not be in figure legends (the authors state: "Details of analysis and statistical data are described in the figure legends.")

3. The Statistical Analysis section of the manuscript could be written far more succinctly. There's a lot of repetition, particularly in the description of the comparison of groups of samples (with paired or unpaired statistical tests). Rather than listing which comparisons were done with which statistical test here, the authors should consider just giving the test name alongside the p-value in brackets in the results section. E.g. ($p = X$; Wilcoxon signed-rank test).

4. For the Bayesian model, the model should be set out first, before explaining how samples were obtained from the model.

5. There is some inaccuracy and imprecision in the description of the statistical analysis and results (in addition to the question of nonparametric T tests raised above):

- "measuring statistical significance" statistical significance is not a measured quantity. "assessing" would be a better term here
- "to quantify uncertainty probabilities" it is not clear what is meant by uncertainty probabilities
- "warm-up" (page 41) should this be burn-in?
- "statistical probability" from legend of Fig 3. This should presumably just be probability (otherwise it is not clear to me what the authors mean by 'statistical probability')
- the acronym FDR in the legend of figure 2 is not given in full anywhere, but presumably refers to false discover rate (even though this is a common acronym it should be expanded on first use). It is not clear what the FDR of 0.1% refers to or how it was calculated. If this is a threshold it should be $FDR < 0.1\%$. The FDR should not be given as a percentage. It also seems unlikely that a FDR of 0.1% corresponded to a p value threshold of 0.001 (these values are the same, except that one is given as a percentage).
- legend of Fig. 5 "To enable comparison across all genes expression values $>6 = 6$ and $<-6 = -6$." This is not sufficiently precise.

6. "Different priors were checked and found to not influence results substantially." More information should be provided on the sensitivity analysis performed and results obtained (as supplementary

material if necessary).

7. Legend of Fig. 2: "T cell responses are shown as log-fold changes based on the baseline input of the complete pMHC library using EdgeR". EdgeR is for gene expression analysis.

It was not clear to me what EdgeR was used for here and there is no mention of the use of EdgeR in the Methods (where a sufficient description should be provided to reproduce the results).

Human endogenous retroviruses form a reservoir of T cell targets in hematological cancers with low mutational burden

REVIEWER COMMENTS

Reviewer #1 (Remarks to the Author):

In their manuscript, Saini SK (sorry for the typo in your name in my initial review) et al described that HERVs form a reservoir of T cell targets in hematological malignancies with low mutational burden. They show that HERV-specific T cells are significantly enriched in patients with myeloid malignancies, and that the presence of such HERV-specific T cells correlated with the expression level of HERVs.

In this revised version of the manuscript, the authors have replied to most of my comments and taken into account my suggestions. This is a huge and serious work now clearly presented and simpler to read. Choice of HERV elements was clarified and explained/illustrated. As I mentioned in my initial revision, I am not a specialist of statistics. Nevertheless it is clear that the authors used in this revised version robust statistical models and suggest interpretations now supported by appropriate statistical evaluation (AG Pedersen's contribution). Figures 3 and 4 were largely reorganized and results presented adequately, and figures 1 and 5 were complemented. References in the text were clarified leading to a better focus of the messages. The discussion section fits better with the presented results.

MINOR COMMENTS

page3 lines 5 to 10:

I would not be so affirmative on the fact that HERV cannot retrotranspose (there are HERV RT activities and LINE RT as well) and the wording "serve" as neutral components is unappropriated. In fact (i) "recently" endogenized retroviral elements are often pathogenic, and HERV that persist after negative selection were either (ii) domesticated to serve beneficial roles (eg Syncytins, some LTR enhancers), or (iii) became neutral components of the host genome. Although there are numerous description of viral particles, the key message is that they are no infectious one (a high number of mutations are required to obtain an infectious strain from "nearly" competent sequences, see Phoenix story, Dewannieux M 2006, and MSR story, Laufer G 2009).

Thank you for your comment and we fully agree with your considerations. Not to confuse the reader, we have now simply removed the statement that the majority of HERVs cannot retrotranspose, but instead serve as neutral components of the human genome, and kept the following:

“The majority of HERVs are defective due to evolutionarily acquired disruption or silencing mutations. Hence, no infectious activity remains from such HERVs, but they may still be recognized as viral components by our immune system.”

page 6 fig 1

There could be some misunderstanding between the 66 initially selected HERVs and the 49 biologically tested HERVs. This would avoid any possible ambiguity if in figure 1 it was explicitly indicated “final library of 1169 peptides from 49 out of 66 HERVs”.

We agree with this consideration and have now added this information in the figure legend (Fig. 1).

In their reply to my comments, the authors justified why they choose to pool all the patients and also PB and BM samples (reply to my “page 7 major comment”, authors points 3 and 4). Obviously, in both situations, it is for sample size reasons/statistical relevance. This nevertheless reveal some limitations of this informative clinical study. A few lines summarizing the limitations of the study and the relevance (and rational) of the choices would be welcome in the discussion section.

Thank you for this comment. We have now added the following sentence to the manuscript’s discussion:

“This was evaluated by combining T cell recognition identified in both PBMCs and BM of the two patient cohorts, as essentially both sampling tissue are part of similar circulating malignant compartment and we didn’t observe any tissue-specific difference in the level of identified T cells. A samples specific comparison between patient and healthy controls would require a larger cohort of patient samples, which was not possible with our sample sets.”

Reviewer #2 (Remarks to the Author):

The authors addressed all points appropriately. Obviously, they could not address the question of cytokines etc. to detect inflammation/T-cell activity in this way, as they do not have/could not obtain serum from the donors. But this does not change the general statement of the paper.

Thank you for this positive feedback.

Reviewer #3 (Remarks to the Author):

We thank the reviewer for providing valuable inputs to improve the statistical evaluation and data representation in our manuscript. We strongly believe that your comments have provided us the opportunity to present the data with clarity on applied statistical methods and uniform representation with proper description of analysis, observation, and results.

Our co-author Prof. Anders Gorm Pedersen has now thoroughly gone over the remaining parts of the manuscript, redone analyses and rewritten sections where appropriate.

The 'Statistical analysis' section has been updated for both better clarity and conciseness. The statistical analysis of the data sets shown in Fig. 4 D-F has been redone using (Bayesian) logistic regression models, summarized in the new Fig. 4D and the new supplementary figure (Fig. S4 C-D).

We hope that you will find the revised manuscript satisfactory for the quality of the statistical analysis, description of methods, and related observations.

Comments for authors

I have reviewed statistical aspects of the manuscript and have the following comments:

1. As pointed out by Reviewer 1 in the initial review the T test is a parametric test. There is no such thing as a nonparametric T test, a Wilcoxon t test or a Mann-Whitney t-test. The fact that this was not understood from the comments of Reviewer 1 is a concern. This error must be fixed everywhere it occurs (it is in the figure legends, the Methods section and perhaps elsewhere).

The reviewer is of course correct: t-tests are not non-parametric, and Mann-Whitney and Wilcoxon tests are not types of t-tests. We regret and apologize for this misunderstanding, which has now been corrected everywhere in the manuscript. The tests are now referred to as Mann–Whitney–Wilcoxon test (instead of Mann-Whitney (t) test), and Wilcoxon signed-rank test (instead of Wilcoxon (t) test).

It is good that the authors have added a bioinformatician as an author on the revised manuscript to take responsibility for the statistical analysis. It is essential, however, that this author should now review and stand over all aspects of the statistical analysis. My impression is that their contribution to the manuscript has been restricted to the Bayesian analysis.

Thank you for this suggestion, all the statistical analyses in the revised manuscript have been redone or checked for appropriateness by Prof. Pedersen.

2. The methods for used for data analysis should be described in sufficient detail in the Methods section and should not be in figure legends (the authors state: "Details of analysis and statistical data are described in the figure legends.")

We have now revised the 'Statistical Analysis' section and included all information related to data analysis in this section or at a relevant place in materials and methods section, and excluded the sentence "Details of analysis and statistical data are described in the figure legend" in the revised version of the manuscript.

3. The Statistical Analysis section of the manuscript could be written far more succinctly. There's a lot of repetition, particularly in the description of the comparison of groups of samples (with paired or unpaired statistical tests). Rather than listing which comparisons were done with which statistical test here, the authors should consider just giving the test name alongside the pvalue in brackets in the results section. E.g. (p = X; Wilcoxon signed-rank test).

Thank you for this suggestion, we have now modified the 'Statistical analysis' accordingly. We have summarized the methods used for the analysis and the p value details and description has now been moved to appropriate figure legends or result section.

4. For the Bayesian model, the model should be set out first, before explaining how samples were obtained from the model.

We have rewritten the entire Statistical analysis with the aim of presenting the material as succinctly and understandably as possible. In this new version, we believe that it reads better to have the sample section before the explanation of individual models (the same sampling scheme was used for all models, so it seemed natural to place it up front).

5. There is some inaccuracy and imprecision in the description of the statistical analysis and results (in addition to the question of nonparametric T tests raised above):
 - "measuring statistical significance" statistical significance is not a measured quantity. "assessing" would be a better term here

The 'statistical analysis' section has been modified and the revised manuscript corrected in terms of the statistical wordings. We have checked for this and similar inaccuracies throughout the manuscript.

- "to quantify uncertainty probabilities" it is not clear what is meant by uncertainty probabilities. This description has been modified in the 'Statistical analysis section, and the revised manuscript does not contain this description.
- "warm-up" (page 41) should this be burn-in?

"Warm-up" and "burn-in" are sometimes used interchangeably, but in this case we have specifically used "warm-up" to emphasize that RStan is tuning the proposal distribution during these steps (while burn-in is mostly used to indicate the initial steps that are thrown

away because the Markov chain was started far from the peak of the posterior, and needs time to reach its equilibrium distribution).

- “statistical probability” from legend of Fig 3. This should presumably just be probability (otherwise it is not clear to me what the authors mean by ‘statistical probability’)

Thank you for pointing out this; we have now changed ‘statistical probability’ to ‘posterior probability’, which is what was actually meant.

- the acronym FDR in the legend of figure 2 is not given in full anywhere, but presumably refers to false discover rate (even though this is a common acronym it should be expanded on first use). It is not clear what the FDR of 0.1% refers to or how it was calculated. If this is a threshold it should be $FDR < 0.1\%$. The FDR should not be given as a percentage. It also seems unlikely that a FDR of 0.1% corresponded to a p value threshold of 0.001 (these values are the same, except that one is given as a percentage).

The FDR refers to false discovery rate. We missed expanding this acronym. To provide more detail on the use of FDR, we have added relevant text under ‘Processing of sequencing data of DNA barcodes’ in the materials and methods section. The FDR was estimated using Benjamini–Hochberg method as described in the original study reporting the use of DNA-barcoded multimers for T cell detection¹, with a threshold of $<0.1\%$ defined as significant for specific enrichment of a given barcodes compared to the full staining pMHC library.

- legend of Fig. 5 “To enable comparison across all genes expression values $>6 = 6$ and $<-6 = -6$.” This is not sufficiently precise.

We have now corrected this:

“For comparative representation, the fold change color scale is restricted from -6 to 6, and fold change values outside this limit are shown at the maximum ($>6 = 6$) or minimum scale ($<-6 = -6$)”

6. “Different priors were checked and found to not influence results substantially.” More information should be provided on the sensitivity analysis performed and results obtained (as supplementary material if necessary).

Thank you for this suggestion, we have now added a plot showing an example of the sensitivity analysis (**Fig. S8**). Specifically, a comparison of estimates using two different priors for the analysis shown in Fig. 3C (the increased level of recognition of HERV-specific T cells in cancer patients, and in post vs pre-AZA treatment). It can be seen that we get essentially identical estimates using either a conservative prior (which tries to keep the estimates close

¹Bentzen, A. K. et al. Large-scale detection of antigen-specific T cells using peptide-MHC-I multimers labeled with DNA barcodes. Nat. Biotechnol. 34, 1037–1045 (2016).

to zero unless the data insist on an effect, and which we used in the paper) or a non-informative prior (which provides very little regularization, and which lets the data sample control the estimate entirely).

7. Legend of Fig. 2: “T cell responses are shown as log-fold changes based on the baseline input of the complete pMHC library using EdgeR”. EdgeR is for gene expression analysis. It was not clear to me what EdgeR was used for here and there is no mention of the use of EdgeR in the Methods (where a sufficient description should be provided to reproduce the results).

We apologize for not providing sufficient details related to data analysis and use of EdgeR package. You rightly mentioned that the EdgeR is designed for gene expression analysis. We have based our analysis on EdgeR and modified for DNA-barcode analysis as described in the original publication².

We have now provided detailed description of this analysis in material and methods section and referred to the publication for better clarity and reproducibility.

²Bentzen, A. K. et al. Large-scale detection of antigen-specific T cells using peptide-MHC-I multimers labeled with DNA barcodes. *Nat. Biotechnol.* 34, 1037–1045 (2016).

REVIEWERS' COMMENTS

Reviewer #3 (Remarks to the Author):

I am satisfied with the revisions to the statistical methods and and method description.